# Online Agnostic Multiclass Boosting

**Vinod Raman**
Department of Statistics
University of Michigan
Ann Arbor, MI 48104
vkraman@umich.edu

**Ambuj Tewari**
Department of Statistics
University of Michigan
Ann Arbor, MI 48018
tewaria@umich.edu

## Abstract

Boosting is a fundamental approach in machine learning that enjoys both strong theoretical and practical guarantees. At a high-level, boosting algorithms cleverly aggregate weak learners to generate predictions with arbitrarily high accuracy. In this way, boosting algorithms convert weak learners into strong ones. Recently, Brukhim et al. [6] extended boosting to the online agnostic binary classification setting. A key ingredient in their approach is a clean and simple reduction to online convex optimization, one that efficiently converts an arbitrary online convex optimizer to an agnostic online booster. In this work, we extend this reduction to multiclass problems and give the first boosting algorithm for online agnostic mutliclass classification. Our reduction also enables the construction of algorithms for statistical agnostic, online realizable, and statistical realizable multiclass boosting.

## 1 Introduction

*Boosting* is a fundamental technique in machine learning that cleverly aggregates the predictions of weak learners to produce a strong learner. Originally studied in the batch (realizable) PAC learning setting for binary classification, boosting has now been extended to a wide variety of settings, including multiclass classification, online learning, and agnostic learning. Recently, Brukhim et al. [6] extended boosting to online agnostic binary classification, marking the completion of boosting algorithms for all four regimes of statistical/online and agnostic/realizable binary classification. However, less can be said about multiclass classification, where boosting algorithms are only well studied under the assumption of realizability. Designing online agnostic multiclass boosting algorithms is important for several reasons. First, realizability is a very strong assumption in real-life: it requires the existence of an expert that *perfectly* labels the data. Second, the vast majority of classification tasks require more than two labels, a prominent example being image classification. Lastly, modern machine learning tasks often require sequential processing of data, making the design and development of online algorithms increasingly relevant. In this work we fill this gap in literature by studying online agnostic boosting for multiclass problems.

### 1.1 Main Results

We give the first weak learning conditions and algorithms for online agnostic multiclass boosting. A key idea of our algorithms is a reduction from boosting to online convex optimization (OCO), an idea borrowed from Brukhim et al. [6]. As a consequence of this reduction, we are also able to give algorithms for the three other settings of statistical agnostic, online realizable, and statistical realizable multiclass boosting. Finally, we give empirical results showcasing that our OCO-based boosting algorithms are fast and competitive with existing state-of-the-art multiclass boosting algorithms.

36th Conference on Neural Information Processing Systems (NeurIPS 2022).

## 1.2 Related Works

Boosting was first studied for binary classification under the realizable PAC learning setting [33, 16–18, 32] and then later extended to the agnostic PAC learning setting [15, 21, 27, 26, 25, 28, 2]. The success of boosting for binary classification led to significant interest in designing boosting algorithms for multiclass problems. As a result, several multiclass boosting algorithms were proposed for the realizable batch setting [19, 14, 22, 20], culminating in the work by Mukherjee and Schapire [30], who unified the previous approaches under a cost matrix framework. Beyond the batch setting, online boosting algorithms for binary [31, 9, 4] and multiclass classification [10, 24] have been designed assuming realizability (mistake-bound). More recently, Brukhim et al. [6] give the first online *agnostic* (regret-bound) boosting algorithm for binary classification, marking complete all four regimes of statistical/online and agnostic/realizable boosting for *binary classification*.

Moving to agnostic *multiclass* boosting, Brukhim et al. [7] study the resources required for boosting in the statistical/batch setting as the number of labels $k$ grows. However, they consider an alternative model of boosting where the weak learner is a strong agnostic PAC learner for a simple "easy-to-learn" base hypothesis class, and the goal is to learn target concepts outside the base class. Specifically, they assume the target concept can be represented by weighted plurality votes over the base class. In this way, the weakness of a weak learner is manifested in the base hypothesis class. Instead, in our work, we consider the standard boosting model of fixing the base hypothesis class, and ask whether a weak learner's performance can be improved relative to the best fixed hypothesis in that class.

Beyond classification, several other works have studied online agnostic boosting for real-valued loss functions under both full-information and bandit feedback settings [3, 1, 5]. In particular, the work by Brukhim and Hazan [5] reduce online boosting for regression tasks under bandit feedback to online linear optimization. A key difference between these works and ours is in the weak learning assumption: these works consider a weak learner that is a strong learner for a small base class of regression functions. The goal of boosting then is to produce a strong online learner for a larger class which contains linear spans of the base class. This is in contrast to this work, where again, we fix the base hypothesis class, and boost the regret bound.

## 2  Preliminaries and Notation

We first describe the basic setup for online agnostic multiclass boosting. There are $k$ possible labels $\mathcal{B}_k := \{1, ..., k\}$ and $k$ is known to the booster and the weak learners. The booster maintains $N$ copies of a weak learner, $\mathcal{W}$, which themselves are (randomized) online learning algorithms that sequentially process examples from instance space $\mathcal{X}$ and output predictions in $\mathcal{B}_k$, which we denote as the set of basis vectors of length $k$. At each iteration $t = 1, ..., T$, an adversary picks a labeled example $(x_t, y_t) \in \mathcal{X} \times \mathcal{B}_k$ and reveals $x_t$ to the booster. Once the booster observes the unlabeled data $x_t$, it gathers the weak learners' predictions and makes a final (possibly randomized) prediction $\hat{y}_t \in \mathcal{B}_k$. After observing the booster's final decision, the adversary reveals the true label $y_t$, and the booster suffers the loss $\mathbb{1}\{\hat{y}_t \neq y_t\}$ or equivalently, *gains* the reward $2\mathbb{1}\{\hat{y}_t = y_t\} - 1$. Finally, the booster, after observing the true label $y_t$, updates each weak learner. Note that the loss/gain of the booster in round $t$ can be written as $1 - \hat{y}_t \cdot y_t$ and $2\hat{y}_t \cdot y_t - 1$ respectively. We also let $\Delta_k$ represent the $(k-1)$-dimensional probability simplex and $\Delta_{\frac{k}{\gamma}}$ represent the $\frac{1}{\gamma}$-scaled $(k-1)$-dimensional probability simplex for $\gamma \in (0, 1)$. Finally, we let $\mathbb{1}_k$ denote the $k$-dimensional ones vector.

**Evaluation**. Unlike the realizable setting, in the agnostic setting, we place no restrictions on how the stream of examples $x_1, ..., x_T$ are labelled. Thus, for a fixed hypothesis class $\mathcal{H} \subseteq \mathcal{B}_k^{\mathcal{X}}$, the goal of the booster is to output predictions $\hat{y}_t$ such that the expected *regret*,

$$\mathbb{E}\left[\max_{h \in \mathcal{H}} \sum_{t=1}^{T} (2h(x_t) \cdot y_t - 1) - \sum_{t=1}^{T} (2\hat{y}_t \cdot y_t - 1)\right],$$

is minimized, where the expectation is over the randomness of the booster and that of the possibly adaptive adversary. Note, this is in contrast to the realizable setting where the stream is labelled by a $h \in \mathcal{H}$ and we wish the booster to minimize the (expected) number of *mistakes* (mistake-bound).

**Agnostic Boosting.** A key technique in agnostic boosting, first appearing in the work by Kanade and Kalai [27], is to update weak learners by feeding randomly *relabelled* examples. This is in contrast to the realizable setting where we typically update weak learners by passing *reweighted* examples.

Accordingly, in order to design a good boosting algorithm, we need to design the booster's strategy for random relabelling while also quantifying the weak learner's ability to maximize cumulative gain, even under relabelled data. The first task will be resolved by allowing the booster to use an Online Convex Optimization (OCO) oracle. In this way, we reduce boosting to OCO, an idea borrowed from Brukhim et al. [6]. For the second task, we give different possible weak learning conditions for the *same* algorithm, all of which capture the ability of a weak learner to maximize cumulative gain with respect to the best fixed competitor in hindsight.

**Online Convex Optimization.** Our booster will use an OCO oracle to update its weak learners. The OCO setting is a sequential game between an online player and adversary over $N$ rounds (see [23] for an in-depth introduction). In each round, the player plays a point $x_i$ in a compact convex set $\mathcal{K} \subset \mathbb{R}^d$, the adversary reveals a loss function $f_i$ chosen from a family of bounded convex functions over $\mathcal{K}$, and the player suffers the loss $f_i(x_i)$. The goal of player is to also minimize *regret*, defined as

$$R(N) = \sum_{i=1}^{N} f_i(x_i) - \min_{x \in \mathcal{K}} \sum_{i=1}^{N} f_i(x).$$

We will denote an algorithm in this setting as $OCO(\mathcal{K}, N)$. If $\mathcal{A}$ is a $OCO(\mathcal{K}, N)$, then we will denote its regret by $R_{\mathcal{A}}(N)$. For many OCO algorithms, like Online Gradient Descent (OGD), the regret $R_{\mathcal{A}}(N)$ is a sub-linear function of the time-horizon $N$.

# 3 Online Agnostic Boosting

In this section, we present an online agnostic boosting algorithm and analyze its performance. We begin in Subsection 3.1 by formally describing our algorithm which reduces boosting to OCO. Then, in Subsection 3.2, we state a weak learning condition and subsequently prove a regret bound for our proposed algorithm.

## 3.1 Algorithm

Pseudocode for our online agnostic boosting algorithm is provided in Algorithm 1. The booster maintains oracle access to $N$ copies of a weak learner $\mathcal{W}_1, ..., \mathcal{W}_N$ as well as a $OCO(\Delta_k, N)$ algorithm $\mathcal{A}$. Each weak learner is characterized by some advantage parameter $\gamma \in (0, 1]$ (directly proportional to its strength), and we will be precise about what exactly $\gamma$ quantifies in Subsection 3.2. In round $t$, the booster uses the weak learners to make a prediction $\hat{y}_t$, observes the true label $y_t$, and finally simulates a game with the OCO algorithm $\mathcal{A}$ to update each weak learner $\mathcal{W}_i$. Specifically, the booster uses the outputs of $\mathcal{A}$ to feed relabelled examples $(x_t, y_t^i)$ to $\mathcal{W}_i$. A critical component of our algorithm is the $L_2$ projection operator onto the probability simplex $\Delta_k$, which we denote by $\prod$. This is used by the booster in line 4 to make randomized predictions $\hat{y}_t$.

We now highlight some desirable properties of Algorithm 1. First, it is easy to implement and efficient assuming access to an efficient weak learner. In particular, for each round $t$, if the running time of the weak learner is $Q$, then the running time of our booster is $O(NQ + Nk \log(k))$. This is in contrast to the OnlineMBBM algorithm proposed by Jung et al. [24] for the realizable setting which is not efficient even assuming access to an efficient weak learner. Second, when $k = 2$, our algorithm reduces down to the online agnostic boosting algorithm proposed by Brukhim et al. [6] in the binary case. Indeed, one can verify that when $k = 2$, the $L_2$ projection operator reduces to the same projection used by Brukhim et al. [6] and the distribution over labels $p_t^i$ induced by the outputs of OCO algorithm are equal for each weak learner in every round $t$.

While the framework in Algorithm 1 is inspired from Brukhim et al. [6] in the binary setting, several new complications arise in the multiclass setting. Most notably, when $k > 2$, we must figure out how the booster should make predictions, what loss functions the booster should construct and pass to $\mathcal{A}$, and lastly, how the booster should use the output of $\mathcal{A}$ to update each weak learner. The interplay behind these three algorithmic pieces is delicate and they have been carefully designed in Algorithm 1 to enable the analysis in Subsection 3.2. Below, we provide some intuition behind these algorithmic decisions.

**Randomized Prediction.** At the start of each round, the booster averages the weak learners votes, scales the average by the parameter $\gamma$, projects the scaled vector back into the simplex, and finally samples a random label. When the $L_2$ projection operator is selected, this approach for randomized

**Algorithm 1:** Online Agnostic Multiclass Boosting via OCO

**Input:** Weak Learners $\mathcal{W}_1...\mathcal{W}_N$, OCO($\Delta_k$, $N$) algorithm $\mathcal{A}$, Advantage parameter $\gamma$

1 **for** $t = 1, ..., T$ **do**
2      Receive example $x_t$
3      Accumulate weak predictions $h_t = \sum_{i=1}^{N} \mathcal{W}_i(x_t)$
4      Set $\mathcal{D}_t = \prod(\frac{h_t}{\gamma N})$
5      Predict $\hat{y}_t \sim \mathcal{D}_t$
6      Receive true label $y_t$
7      **for** $i = 1, ..., N$ **do**
8          If $i > 1$, obtain $p_t^i = \mathcal{A}(l_t^1, ..., l_t^{i-1})$. Else, initialize $p_t^1 = \frac{\mathbb{1}_k}{k}$.
9          Reveal loss function: $l_t^i(p) = p \cdot \left( \frac{2\mathcal{W}_i(x_t) - \mathbb{1}_k}{\gamma} - (2y_t - \mathbb{1}_k) \right)$
10          Sample random label $y_t^i \sim p_t^i$, and pass $(x_t, y_t^i)$ to $\mathcal{W}_i$
11      Reset $\mathcal{A}$

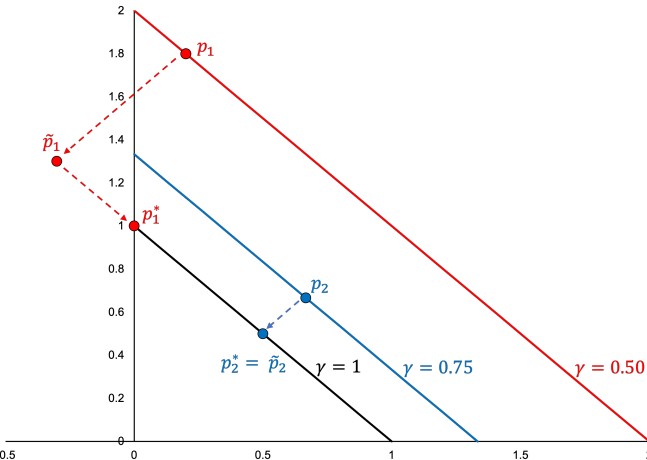

Figure 1: The red, blue, and black lines correspond to $\frac{\Delta_2}{\gamma}$ for $\gamma = 0.50, 0.75$, and, $1.0$ respectively. $p_1$ and $p_2$ are $\gamma$-scaled votes and $p_1^*$ and $p_2^*$ are their corresponding $L_2$ projection onto $\Delta_2$. $\tilde{p}_1$ and $\tilde{p}_2$ denote the orthogonal projections onto the plane containing $\Delta_2$.

prediction achieves a *polarization* effect: as $\gamma$ gets smaller, the projection concentrates mass on a fewer number of labels, specifically those labels that have achieved the majority of the votes from the weak learners. When $\gamma$ nears 0, the $L_2$ projection eventually places all mass on the label with the most votes.

One might think that the $L_2$ projection is not a natural projection operator for $\Delta_k$ (as KL or $L_1$ projections might seem better suited to the geometry of the probability simplex). However, in the case where we project from $\Delta_{\frac{k}{\gamma}}$ to $\Delta_k$, we find that it is a natural choice from a geometric perspective. Figure 1 provides a visualization of the $L_2$ projection operator from $\Delta_{\frac{k}{\gamma}}$ to $\Delta_k$ for $k = 2$. Because the spaces $\Delta_{\frac{k}{\gamma}}$ and $\Delta_k$ are parallel, given any point $p \in \Delta_{\frac{k}{\gamma}}$, one can think of its $L_2$ projection in the following procedural manner:

     1. Compute the *orthogonal* projection, $\tilde{p}$, onto the plane containing $\Delta_k$.
     2. If $\tilde{p} \in \Delta_k$, output $p^* = \tilde{p}$.
     3. Else, output $p^* = \arg\min_{p \in \Delta_k} ||p - \tilde{p}||_2$.

From this procedural perspective, Figure 1 lends a geometric intuition behind the polarization effect of the $L_2$ projection. As $\gamma$ shrinks, the number of points in $\Delta_{\frac{k}{\gamma}}$ that lie orthogonally above $\Delta_k$ shrinks. Thus, the orthogonal projection of the vast majority of points in $\Delta_{\frac{k}{\gamma}}$ will lie outside $\Delta_k$,

leading to a greater number of sparse projections that lie on the boundary of $\Delta_k$. From the booster's viewpoint, this property of the $L_2$ projection is desirable. If the weak learners are very weak (small $\gamma$) but somehow concentrate votes on a few labels, then it may be likely that the true response is amongst these few labels. In this sense, $\gamma$ controls how many weak learners need to agree on a particular label to convince the booster to deterministically predict that label.

**Updating Weak Learners**. Once the true label $y_t$ is revealed in each round, the booster must update each weak learner. As mentioned in Section 2, one strategy for updating weak learners in the agnostic setting is via *random relabelling*. Indeed, in line 10, the booster passes back to the weak learner the example $x_t$ with a random label $y_t^i \sim p_t^i$. Together, the specified loss function in line 9 and random relabelling strategy in line 10 achieve the following effect: if more weak learners make mistakes, the distribution over labels output by the OCO algorithm $\mathcal{A}$ in line 8 concentrates on the true label $y_t$, increasing the likelihood that $y_t$ is passed to subsequent weak learners. This is desirable as the outputs of the OCO guide each weak learner to correct for the mistakes of preceding learners.

## 3.2 Regret Analysis

Before we give the regret bound of Algorithm 1, we need to specify the capacity to which the weak learners can make predictions, even under potentially relabelled data. Unfortunately, in the agnostic setting, there is no canonical weak learning condition for multiclass problems. In this paper, we give *several* possible definitions of an Agnostic Weak Online Learner (AWOL) for the multiclass setting, all of which enable a regret analysis of Algorithm 1. We emphasize that we can derive different regret bounds for Algorithm 1 based on what condition we assume our weak learners to satisfy. For this section, we present a weak learning condition based loosely on a *one-vs-one* approach to multiclass classification. In Appendix C, we provide alternative weak learning conditions.

Define the gain function for input $z \in \mathcal{B}_k$,

$$\sigma_{y,\ell}(z) = \mathbb{1}\{z = y\} - \mathbb{1}\{z = \ell\} = z \cdot (y - \ell).$$

For some $y \in \mathcal{B}_k$ and $\ell \in \mathcal{B}_k \setminus \{y\}$, $\sigma_{y,\ell}(\cdot)$ can be thought of as the binary classification task between labels $y$ and $\ell$. Definition 1 requires that for any such sequence of binary classification tasks, an online agnostic weak learner must be able to eventually distinguish between every pair of labels to some non-trivial, but far from optimal, degree.

**Definition 1** (Agnostic Weak Online Learning). *Let $\mathcal{H} \subseteq \mathcal{B}_k^{\mathcal{X}}$ be a class of experts and let $0 < \gamma \leq 1$ denote the "advantage". An online learning algorithm $\mathcal{W}$ is a $(\gamma, T)$-agnostic weak online learner (AWOL) for $\mathcal{H}$ if for any adaptively chosen sequence of tuples $(x_t, y_t, \ell_t) \in \mathcal{X} \times \mathcal{B}_k \times \mathcal{B}_k$ where $\ell_t \neq y_t$, the algorithm outputs $\mathcal{W}(x_t) \in \mathcal{B}_k$ at every iteration $t \in [T]$ such that,*

$$\gamma \max_{h \in \mathcal{H}} \mathbb{E}\left[\sum_{t=1}^{T} \sigma_{y_t,\ell_t}(h(x_t))\right] - \mathbb{E}\left[\sum_{t=1}^{T} \sigma_{y_t,\ell_t}(\mathcal{W}(x_t))\right] \leq R_{\mathcal{W}}(T, k),$$

*where the expectation is taken w.r.t. the randomness of the weak learner $\mathcal{W}$ and that of the possibly adaptive adversary, and $R_{\mathcal{W}} : \mathbb{N} \times \mathbb{N} \to \mathbb{R}_+$ is the additive regret: a non-decreasing, sub-linear function of $T$.*

The precise dependence of $R_{\mathcal{W}}(T, k)$ on $k$ is explored in more detail in Appendix D where we explicitly construct learners satisfying AWOL. We make few remarks about Definition 1. First, the strength of the weak learner varies directly with $\gamma$. Second, for $k = 2$, Definition 1 reduces to the weak learning condition by Brukhim et al. [6] in the binary setting. Finally, we emphasize that Definition 1 holds under an *adaptive* adversary, one that can choose $(y_t, \ell_t)$ based on $\{\mathcal{W}(x_i)\}_{i=1}^{t-1}$ and its own internal random bits. Importantly, we also allow the adversary to pick $\ell_t$ even after it has observed $\mathcal{W}(x_t)$ as this only controls how much loss the weak learner suffers when it is incorrect.

Under the assumption that weak learners satisfy Definition 1, Theorem 1 bounds the regret of Algorithm 1 under an *oblivious* adversary. Using a standard reduction, our results can then be generalized to an *adaptive* adversary (see Chapter 4 in [8]). In particular, a key requirement allowing an oblivious regret bound to generalize to an adaptive regret bound is that the learner's predictions on round $t$ should not depend on any of its past predictions from previous rounds. This is indeed true for our Booster.

**Theorem 1** (Regret Bound). *Assuming weak learners satisfy Definition 1, the expected regret bound of Algorithm 1 is,*

$$\frac{1}{T}\,\mathbb{E}\left[\max_{h\in\mathcal{H}}\sum_{t=1}^{T}\left(2h(x_t)\cdot y_t - 1\right) - \sum_{t=1}^{T}(2\hat{y}_t\cdot y_t - 1)\right] \leq \frac{R_{\mathcal{W}}(T,k)}{\gamma T} + \frac{R_{\mathcal{A}}(N)}{N},$$

*where the expectation is over the randomness of the algorithm and weak learners, and $R_{\mathcal{W}}(T,k)$, $R_{\mathcal{A}}(N)$ are the regret terms of the AWOL, OCO algorithms respectively.*

If one picks $\mathcal{A}$ to be *Online Gradient Descent* (OGD), then $R_{\mathcal{A}}(N) = O(GD\sqrt{N})$, where $D$ is the diameter of $\Delta_k$ and $G$ is the upper-bound on $||\nabla_p l_t^i(p)||$. In our setting, $D = \sqrt{2}$ and $G = O(\frac{1}{\gamma})$ (using Lemma 19) and hence the average regret further simplifies to:

$$\frac{1}{T}\,\mathbb{E}\left[\max_{h\in\mathcal{H}}\sum_{t=1}^{T}\left(2h(x_t)\cdot y_t - 1\right) - \sum_{t=1}^{T}(2\hat{y}_t\cdot y_t - 1)\right] \leq \frac{R_{\mathcal{W}}(T,k)}{\gamma T} + O\left(\frac{1}{\gamma\sqrt{N}}\right).$$

To exemplify the role of $N$, consider a scenario where $R_{\mathcal{W}}(T,k) = O(k\sqrt{T})$ (see Appendix D for examples). By setting $N = \frac{T}{\gamma^2}$ the overall regret of the booster becomes $O(\frac{k\sqrt{T}}{\gamma})$. In the next subsection, we give the proof of Theorem 1.

### 3.2.1 Proof of Theorem 1

We follow a similar procedure to Brukhim et al. [6] by lower and upper bounding the expected sum of losses passed to $\mathcal{A}$ in terms of the regret of the weak learner and the regret of $\mathcal{A}$ respectively. These bounds rely on several lemmas that have been abstracted out and provided in Appendix E. As mentioned previously, we also assume an oblivious adversary.

Let $(x_1, y_1), ..., (x_T, y_T)$ be any sequence of example-label pairs. We start by giving a lower bound on the expected sum of losses passed to $\mathcal{A}$ using Definition 1. Define $h^* = \arg\max_{h\in\mathcal{H}}\sum_{t=1}^{T}(2h(x_t)\cdot y_t - 1)$ as the optimal competitor in hindsight and let

$$\ell_t^i = \begin{cases} \mathcal{W}_i(x_t), & \text{if } \mathcal{W}_i(x_t) \neq y_t^i \\ \ell \in \mathcal{B}_k \setminus \{y_t^i\}, & \text{otherwise} \end{cases}.$$

Note that the precise choice of $\ell_t^i$ in the second case is not important. The proof below only uses the fact that $\ell_t^i \in \mathcal{B}_k \setminus \{y_t^i\}$ in that case. We then have,

$$\mathbb{E}\left[\sum_{i=1}^{N}\sum_{t=1}^{T}l_t^i(p_t^i)\right] = \mathbb{E}\left[\sum_{i=1}^{N}\sum_{t=1}^{T}p_t^i\cdot\left(\frac{2\mathcal{W}_i(x_t)-\mathbb{1}_k}{\gamma} - (2y_t - \mathbb{1}_k)\right)\right]$$

$$= \frac{1}{\gamma}\sum_{i=1}^{N}\sum_{t=1}^{T}\mathbb{E}\left[p_t^i\cdot(2\mathcal{W}_i(x_t)-\mathbb{1}_k)\right] - \sum_{i=1}^{N}\sum_{t=1}^{T}\mathbb{E}\left[p_t^i\cdot(2y_t - \mathbb{1}_k)\right]$$

$$= \frac{1}{\gamma}\sum_{i=1}^{N}\sum_{t=1}^{T}\mathbb{E}\left[2\mathcal{W}_i(x_t)\cdot y_t^i - 1\right] - \sum_{i=1}^{N}\sum_{t=1}^{T}\mathbb{E}\left[p_t^i\cdot(2y_t - \mathbb{1}_k)\right] \qquad \text{(Lemma 15)}$$

$$= \frac{1}{\gamma}\sum_{i=1}^{N}\sum_{t=1}^{T}\mathbb{E}\left[\sigma_{y_t^i,\ell_t^i}(\mathcal{W}_i(x_t))\right] - \sum_{i=1}^{N}\sum_{t=1}^{T}\mathbb{E}\left[p_t^i\cdot(2y_t - \mathbb{1}_k)\right].$$

Using the weak learning condition in Definition 1,

$$\frac{1}{\gamma} \sum_{i=1}^{N} \sum_{t=1}^{T} \mathbb{E}\left[\sigma_{y_t^i,\ell_t^i}(\mathcal{W}_i(x_t))\right] \geq \sum_{i=1}^{N} \max_{h \in \mathcal{H}} \sum_{t=1}^{T} \mathbb{E}\left[\sigma_{y_t^i,\ell_t^i}(h(x_t))\right] - \frac{NR_{\mathcal{W}}(T,k)}{\gamma} \qquad \text{(Definition 1)}$$

$$\geq \sum_{i=1}^{N} \sum_{t=1}^{T} \mathbb{E}\left[\sigma_{y_t^i,\ell_t^i}(h^*(x_t))\right] - \frac{NR_{\mathcal{W}}(T,k)}{\gamma}$$

$$\geq \sum_{i=1}^{N} \sum_{t=1}^{T} \mathbb{E}\left[2h^*(x_t) \cdot y_t^i - 1\right] - \frac{NR_{\mathcal{W}}(T,k)}{\gamma}$$

$$= \sum_{i=1}^{N} \sum_{t=1}^{T} \mathbb{E}\left[p_t^i \cdot (2h^*(x_t) - \mathbb{1}_k)\right] - \frac{NR_{\mathcal{W}}(T,k)}{\gamma}. \qquad \text{(Lemma 15)}$$

Putting things together, we find,

$$\mathbb{E}\left[\sum_{i=1}^{N} \sum_{t=1}^{T} l_t^i(p_t^i)\right] \geq \sum_{i=1}^{N} \sum_{t=1}^{T} \mathbb{E}\left[p_t^i \cdot (2h^*(x_t) - 2y_t)\right] - \frac{NR_{\mathcal{W}}(T,k)}{\gamma}$$

$$\geq \sum_{i=1}^{N} \sum_{t=1}^{T} \mathbb{E}\left[2(h^*(x_t) \cdot y_t - 1)\right] - \frac{NR_{\mathcal{W}}(T,k)}{\gamma} \qquad \text{(Lemma 16)}$$

$$= N \sum_{t=1}^{T} 2(h^*(x_t) \cdot y_t - 1) - \frac{NR_{\mathcal{W}}(T,k)}{\gamma}.$$

Now, we compute an upper bound. For any $t \in [T]$ and arbitrary $p^* \in \Delta_k$:

$$\mathbb{E}\left[\frac{1}{N} \sum_{i=1}^{N} l_t^i(p_t^i)\right] \leq \frac{1}{N} \mathbb{E}\left[\min_{p \in \mathcal{K}} \sum_{i=1}^{N} l_t^i(p)\right] + \frac{R_{\mathcal{A}}(N)}{N} \qquad \text{(OCO Regret)}$$

$$\leq \frac{1}{N} \mathbb{E}\left[\sum_{i=1}^{N} l_t^i(p^*)\right] + \frac{R_{\mathcal{A}}(N)}{N}$$

$$= \frac{1}{N} \sum_{i=1}^{N} \mathbb{E}\left[p^* \cdot \left(\frac{2\mathcal{W}_i(x_t) - \mathbb{1}_k}{\gamma} - (2y_t - \mathbb{1}_k)\right)\right] + \frac{R_{\mathcal{A}}(N)}{N}$$

$$= \frac{1}{\gamma N} \sum_{i=1}^{N} \left(p^* \cdot \mathbb{E}\left[2\mathcal{W}_i(x_t) - \mathbb{1}_k\right]\right) - p^* \cdot (2y_t - \mathbb{1}_k) + \frac{R_{\mathcal{A}}(N)}{N}$$

$$= p^* \cdot \left(\frac{1}{\gamma N} \sum_{i=1}^{N} \mathbb{E}\left[2\mathcal{W}_i(x_t) - \mathbb{1}_k\right]\right) - p^* \cdot (2y_t - \mathbb{1}_k) + \frac{R_{\mathcal{A}}(N)}{N}$$

$$= \mathbb{E}\left[p^* \cdot \left(\frac{\frac{2}{N}\sum_{i=1}^{N} \mathcal{W}_i(x_t) - \mathbb{1}_k}{\gamma} - (2y_t - \mathbb{1}_k)\right)\right] + \frac{R_{\mathcal{A}}(N)}{N}$$

$$\leq \mathbb{E}\left[2\left(\prod\left(\frac{1}{\gamma N} \sum_{i=1}^{N} \mathcal{W}_i(x_t)\right) \cdot y_t - 1\right)\right] + \frac{R_{\mathcal{A}}(N)}{N} \qquad \text{(Lemma 17)}$$

$$= 2\left(\mathbb{E}\left[\hat{y}_t\right] \cdot y_t - 1\right) + \frac{R_{\mathcal{A}}(N)}{N}. \qquad \text{(Law of total expectation)}$$

Summing over $T$,

$$\mathbb{E}\left[\frac{1}{N} \sum_{t=1}^{T} \sum_{i=1}^{N} l_t^i(p_t^i)\right] \leq \sum_{t=1}^{T} 2\left(\mathbb{E}\left[\hat{y}_t\right] \cdot y_t - 1\right) + \frac{TR_{\mathcal{A}}(N)}{N}.$$

Combining lower and upper bounds for $\mathbb{E}\left[\frac{1}{NT}\sum_{i=1}^{N}\ell_t^i(p_t^i)\right]$, we get

$$\frac{1}{T}\mathbb{E}\left[\max_{h\in\mathcal{H}}\sum_{t=1}^{T}(2h(x_t)\cdot y_t - 1) - \sum_{t=1}^{T}(2\hat{y}_t\cdot y_t - 1)\right] \leq \frac{R_{\mathcal{W}}(T,k)}{\gamma T} + \frac{R_{\mathcal{A}}(N)}{N},$$

which completes the proof. Since $2h(x_t)\cdot y_t - 1 = 1 - 2\mathbb{1}_k\{h(x_t)\neq y_t\}$, we can also write the average regret in terms of *mistakes*,

$$\frac{1}{T}\mathbb{E}\left[\sum_{t=1}^{T}\mathbb{1}\{\hat{y}_t\neq y_t\} - \min_{h\in\mathcal{H}}\sum_{t=1}^{T}\mathbb{1}\{h(x_t)\neq y_t\}\right] \leq \frac{R_{\mathcal{W}}(T,k)}{2\gamma T} + \frac{R_{\mathcal{A}}(N)}{2N}.$$

# 4 Beyond Online Agnostic Boosting

In this section, we present results of extending our reduction to the three other boosting settings, namely statistical agnostic, online realizable, and statistical realizable learning. The purpose of this section is to showcase the generality of the OCO-based boosting framework and not to achieve state-of-the-art bounds for these settings. Boosting algorithms and all associated proofs can be found in Appendix A and B. Throughout this section we will let $\mathcal{W}_S$ denote the hypothesis output by a weak learner trained on a sample $S$.

## 4.1 Statistical Agnostic Boosting

In the statistical setting, our objective of interest is the *correlation*, which we define below. Let $\mathcal{D}$ be a distribution over $\mathcal{X}\times\mathcal{B}_k$ and let $h:\mathcal{X}\to\mathcal{B}_k$ be an hypothesis. Define the *multiclass* correlation of $h$ with respect to $\mathcal{D}$ as

$$cor_D(h) = \mathbb{E}_{(x,y)\sim D}\left[h(x)\cdot(2y-\mathbb{1}_k)\right].$$

Like the online agnostic setting, the boosting algorithm for this setting (provided in Appendix A) can be analyzed under several candidate weak learning conditions. To showcase the dependence on $k$, we provide a weak learning condition based loosely on a *one-vs-all* approach to multiclass classification.

**Definition 2** (Empirical Agnostic Weak Learning). *Let $\mathcal{H}\subseteq\mathcal{B}_k^{\mathcal{X}}$ be a hypothesis class and let $0<\gamma\leq 1$ denote the "advantage". Let $\boldsymbol{x}=(x_1,...,x_m)\in\mathcal{X}$ denote an unlabeled sample. A learning algorithm $\mathcal{W}$ is a $(\gamma,\epsilon_0,m_0)$-agnostic weak learner (AWL) for $\mathcal{H}$ with respect to $\boldsymbol{x}$ if for any labels $\boldsymbol{y}=(y_1,...,y_m)\in\mathcal{B}_k$, and every reference label $\ell\in\mathcal{B}_k$,*

$$\mathbb{E}_{S'}\left[\sum_{i:y_i=\ell}\mathcal{W}_{S'}(x_i)\cdot(2y_i-\mathbb{1}_k)\right] \geq \gamma\max_{h\in\mathcal{H}}\sum_{i:y_i=\ell}h(x_i)\cdot(2y_i-\mathbb{1}_k) - m\epsilon_0,$$

*where $S'$ is an independent sample of size $m_0$ drawn from the distribution which uniformly assigns to each example $(x_i,y_i)$ probability $1/m$.*

Under the assumption that our weak learners satisfy Definition 2, Corollary 2 bounds the expected correlation of a statistical agnostic boosting algorithm.

**Corollary 2** (Empirical Agnostic Correlation Bound). *There exists a boosting algorithm whose output after $T$ rounds, denoted by $\bar{h}$, satisfies:*

$$\mathbb{E}\left[cor_S(\bar{h})\right] \geq \max_{h\in\mathcal{H}}\mathbb{E}\left[cor_S(h)\right] - \frac{R_{\mathcal{A}}(T)}{T} - \frac{k\epsilon_0}{\gamma},$$

*where $S$ is the distribution which uniformly assigns to each example $(x_i,y_i)$ probability $1/m$.*

Letting $\mathcal{A}$ be OGD, and setting $T=O(\frac{1}{\gamma^2\epsilon^2})$ for any $\epsilon>0$ gives an error rate of $\frac{k\epsilon_0}{\gamma}+\epsilon$. Although, to our best knowledge, there are no existing multiclass boosting algorithms in the agnostic setting, we point out that there is evidence to suggest that a sub-optimal dependence of the error rate on $k$ might be unavoidable (see [7]).

## 4.2 Online Realizable Boosting

In the online realizable setting, we are guaranteed that the stream of examples $(x_1, y_1), ..., (x_T, y_T)$ is perfectly labelled by some hypothesis $h \in \mathcal{H}$. Definition 3 then gives an appropriate online realizable weak learning condition.

**Definition 3** (Realizable Weak Online Learning). *Let $\mathcal{H} \subseteq \mathcal{B}_k^{\mathcal{X}}$ be a class of experts and let $0 < \gamma \leq 1$ denote the "advantage". An online learning algorithm $\mathcal{W}$ is a $(\gamma, T)$-realizable weak online learner (**RWOL**) for $\mathcal{H}$ if for any sequence $(x_1, y_1), ..., (x_T, y_T) \in \mathcal{X} \times \mathcal{B}_k$ that is realizable by $\mathcal{H}$, at every iteration $t \in [T]$, the algorithm outputs $\mathcal{W}(x_t) \in \mathcal{B}_k$ such that,*

$$\mathbb{E}\left[\sum_{t=1}^{T}(2\mathcal{W}(x_t) \cdot y_t - 1)\right] \geq \gamma T - R_{\mathcal{W}}(T, k),$$

*where the expectation is taken w.r.t. the randomness of the weak learner $\mathcal{W}$ and $R_{\mathcal{W}} : \mathbb{N} \times \mathbb{N} \to \mathbb{R}_+$ is the additive regret: a non-decreasing, sub-linear function of $T$.*

Under Definition 3, Corollary 3 bounds the expected gain of an online boosting algorithm.

**Corollary 3** (Mistake Bound). *There exists a boosting algorithm whose outputs $\hat{y}_1, ..., \hat{y}_T$ satisfy:*

$$\frac{1}{T}\sum_{t=1}^{T}(2\,\mathbb{E}\left[\hat{y}_t\right] \cdot y_t - 1) \geq 1 - \frac{R_{\mathcal{A}}(N)}{N} - \frac{\tilde{R}_{\mathcal{W}}(T, k)}{\gamma T},$$

*where $\tilde{R}_{\mathcal{W}}(T, k) = 2R_{\mathcal{W}}(T, k) + \tilde{O}(\sqrt{T})$.*

If we consider a scenario where $\tilde{R}_{\mathcal{W}}(T, k) = O(k\sqrt{T})$, taking $\mathcal{A}$ to be OGD, $N$ to be $O(\frac{1}{\gamma^2\epsilon^2})$ and $T$ to be $O(\frac{1}{\gamma^2\epsilon^2})$, gives a mistake-bound at most $\epsilon kT$. We point out that we cannot readily compare this bound with the existing bounds for online realizable boosting by Jung et al. [24] because the dependence of $R_{\mathcal{W}(T,k)}$ on $k$ depends on the weak learner of choice.

## 4.3 Statistical Realizable Boosting

In the statistical realizable setting, our metric of interest is again the multiclass correlation. However, under the realizability assumption, $\max_{h^* \in \mathcal{H}} cor_D(h^*) = 1$. Accordingly, Definition 4 gives an appropriate realizable weak learning condition.

**Definition 4** (Empirical Realizable Weak Learning). *Let $\mathcal{H} \subseteq \mathcal{B}_k^{\mathcal{X}}$ be a hypothesis class and let $0 < \gamma \leq 1$ denote the "advantage". Let $S = \{(x_1, y_1), ..., (x_m, y_m)\} \in \mathcal{X} \times \mathcal{B}_k$ be a sample. A learning algorithm $\mathcal{W}$ is a $(\gamma, m_0)$ - realizable weak learner (**RWL**) for $\mathcal{H}$ with respect to $S$ if for any distribution $\boldsymbol{p} = (p_1, ..., p_m)$ over the examples,*

$$\mathbb{E}_{S'}[cor_{\boldsymbol{p}}(\mathcal{W}_{S'})] \geq \gamma,$$

*where $S'$ is an independent sample of size $m_0$ drawn from $\boldsymbol{p}$.*

Under Definition 4, Corollary 4 bounds the expected correlation of a statistical realizable boosting algorithm.

**Corollary 4** (Empirical Correlation Bound). *There exists a boosting algorithm whose output after $T$ rounds, denoted $\bar{h}$, satisfies,*

$$\mathbb{E}\left[cor_S(\bar{h})\right] \geq 1 - \frac{R_{\mathcal{A}}(T)}{T},$$

*where $S$ is the distribution which uniformly assigns to each example $(x_i, y_i)$ probability $1/m$.*

Note that the cost of weak learning manifests in the regret term of the OCO algorithm. Precisely, if one picks OGD to be the OCO algorithm, then the bound on correlation can be expressed as $\mathbb{E}\left[cor_S(\bar{h})\right] \geq 1 - O\left(\frac{1}{\gamma\sqrt{T}}\right)$, which exhibits a decreasing lower bound as $\gamma$ shrinks. Setting $T = O(\frac{1}{\gamma^2\epsilon^2})$ for any $\epsilon > 0$ ensures that at most $\epsilon$ error is obtained. It is difficult to compare our error rates to that achieved by existing multiclass boosting algorithm in the realizable setting because our weak learning condition is different and potentially stronger. Nevertheless, our bounds are sub-optimal to those in Mukherjee and Schapire [30] which only require setting $T = O(\frac{1}{\gamma^2}\log(\frac{k}{\epsilon}))$ to achieve $\epsilon$ error.

Table 1: Average accuracy and runtime of algorithms on 7 UCI datasets.

| Dataset | $k$ | Accuracy(%) | | | | Runtime(s) | | | |
|---|---|---|---|---|---|---|---|---|---|
| | | Agn | Opt | Ada | OCOR | Agn | Opt | Ada | OCOR |
| Balance | 3 | **84.4** | 79.5 | 75.2 | 78.8 | 11.5 | 57.6 | 7.1 | 4.5 |
| Cars | 4 | **80.6** | 69.9 | 75.8 | 69.6 | 42.5 | 104.1 | 20.4 | 12.1 |
| Landsat | 6 | 67.0 | **80.8** | 56.9 | 79.6 | 1255.6 | 1814.3 | 81.8 | 440.4 |
| Segment | 7 | 75.0 | 78.9 | 68.6 | **79.3** | 647.2 | 2448.3 | 51.8 | 154.7 |
| Mice | 8 | **86.0** | 77.6 | 71.3 | 79.6 | 1025.8 | 1811.6 | 90.1 | 258.2 |
| Yeast | 10 | 42.3 | 39.8 | 41.7 | **47.6** | 348.3 | 1468.8 | 25.2 | 56.6 |
| Abalone | 28 | 22.1 | **24.9** | 19.2 | 22.0 | 4036.7 | 9775.3 | 108.0 | 248.2 |

## 5 Experiments

We performed experiments with Algorithm 1 on seven UCI datasets [13]. Since, to our knowledge, there are no known online *agnostic* boosting algorithms for multiclass problems, we benchmark performance against three online *realizable* multiclass boosting algorithms: the state-of-the-art optimal (OnlineMBBM) and adaptive (Adaboost.OLM) online boosting algorithms by Jung et al. [24], and the OCO-based online boosting algorithm alluded to in Section 4 (see Appendix B). For weak learners, we used the implementation of the VeryFastDecisionTree from the River package [29] and restricted the maximum depth of the tree to 1. We used Projected OGD [35] for the OCO algorithm and set the number of weak learners, $N$, to 100 for each boosting algorithm.

Our experiments using Algorithm 1 are performed using *fractional relabeling*, a technique borrowed from Kanade and Kalai [27]. That is, instead of passing just a single example-label pair $(x_t, y_t^i)$ where $y_t^i \sim p_t^i$, we pass all $k$ possible *weighted* example-label tuples $\{(x_t, \ell, p_t^i[\ell])\}_{\ell=1}^k$ to weak learner $i$ in round $t$. Experiments with random relabeling showed that random relabeling runs faster but performs worse than fractional relabeling.

Table 1 summarizes the average accuracy and runtime over five independent shuffles of each dataset for each boosting algorithm. "Agn" refers to Algorithm 1, "Opt" to OnlineMBBM, "Ada" to Adaboost.OLM, and "OCOR" to the OCO-based online realizable algorithm. Standard errors for accuracies are provided in Appendix F. All algorithms **except** AdaBoost.OLM are parameterized by an advantage parameter $\gamma \in (0, 1)$. Thus, $\gamma$ was tuned separately for each respective cell of Table 1. See Appendix F for more experimental details. All code is available at `https://github.com/vinodkraman/OnlineAgnosticMulticlassBoosting`.

Despite having the fastest overall runtime, Adaboost.OLM had the poorest performance. Compared to OnlineMBBM, our OCO-based boosting algorithms achieve comparable performance at a fraction of the runtime. Specifically, for three of our datasets, our online agnostic boosting algorithm achieves the highest accuracy. For the remaining datasets, our OCO-based realizable boosting algorithm achieves comparable accuracy to OnlineMBBM with shorter runtimes.

## 6 Discussion

We give the first weak learning conditions and algorithm for online agnostic multiclass boosting. Our algorithm relies on a clean and simple reduction from boosting to online convex optimization. This fruitful connection allows us to go beyond the online agnostic setting and design multiclass boosting algorithms for all four regimes of statistical/online and agnostic/realizable learning. As future work, we leave it open to identify the correct weak learning condition, construct adaptive versions of our boosting algorithms, improve regret upper bounds/prove lower bounds, design agnostic boosting algorithms under bandit feedback, and study the impact of the choice of weak learner and OCO algorithm on the empirical performance of our OCO-based boosting algorithms.

**Acknowledgements.** AT acknowledges the support of NSF via grant IIS-2007055.

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
