# A Statistical Agnostic Boosting

We fill in the gaps of Section 4 for the statistical agnostic setting by providing a boosting algorithm and its theoretical analysis.

## A.1 Algorithm

We now describe a statistical agnostic boosting algorithm whose pseudocode is provided below. The booster is given as input a sample $S = (x_1, y_1), ..., (x_m, y_m) \in \mathcal{X} \times \mathcal{B}_k$. The booster has black-box oracle access to $T$ copies of a $(\gamma, \epsilon_0, m_0)$-AWL algorithm, $\mathcal{W}_1, ..., \mathcal{W}_T$, each satisfying Definition 2, and $m$ copies of an OCO$(\Delta_k, T)$ algorithm, $\mathcal{A}^1, ..., \mathcal{A}^m$. Importantly, note that in line 8 , we denote by $\prod$ as a randomized *prediction* operator that given a $\gamma$-scaled distribution over $k$ labels, first computes the $L_2$ projection onto $\Delta_k$, and then randomly samples a label from the resulting distribution.

---

**Algorithm 2:** Statistical Agnostic Multiclass Boosting via OCO

---

**Input:** $(\gamma, \epsilon_0, m_0)$-AWLs $\mathcal{W}_1...\mathcal{W}_T$, OCO$(\Delta_k, T)$ algorithms $\mathcal{A}^1...\mathcal{A}^m$, and Sample
$\quad\quad S = (x_1, y_1)...(x_m, y_m) \in \mathcal{X} \times \mathcal{B}_k$

1 **for** $t = 1, ..., T$ **do**

2 $\quad$ If $t = 1$, set $P_1[i] = \frac{\mathbb{1}_k}{k} \ \forall i \in [m]$

3 $\quad$ Pass $m_0$ examples to $\mathcal{W}_t$ drawn from the following distribution:

4 $\quad$ Draw $x_i$ w.p. $\frac{1}{m}$ and relabel by drawing $\tilde{y}_i \sim P_t[i]$

5 $\quad$ Let $h_t$ be the weak hypothesis returned by $\mathcal{W}_t$

6 $\quad$ Reveal loss function: $l_t^i(p) = (2p - \mathbb{1}_k) \cdot \left( \frac{h_t(x_i)}{\gamma} - y_i \right) \ \forall i \in [m]$

7 $\quad$ Set $P_{t+1}[i] = \mathcal{A}^i(l_1^i, ..., l_t^i) \ \forall i \in [m]$

8 **return** $\bar{h}(x) = \prod \left( \frac{1}{\gamma T} \sum_{t=1}^{T} h_t(x) \right)$

---

Once again, the high-level framework is inspired by the work of Brukhim et al. [6], but like in the online setting, several pieces need to be redesigned when $k > 2$. One key difference between our algorithm and the corresponding algorithm by Brukhim et al. [6] is the use of multiple OCO algorithms. The algorithmic choices in Algorithm 2 are similar to those made in the online setting. Thus, the intuition provided in Section 3.1 also follows here.

Under the assumption that the weak learners satisfy Definition 2, Theorem 5 bounds the *correlation* of Algorithm 2.

**Theorem 5.** *The output of Algorithm 2, which is denoted $\bar{h}$, satisfies,*

$$\mathbb{E}\left[cor_S(\bar{h})\right] \geq \max_{h \in \mathcal{H}} cor_S(h) - \frac{R_{\mathcal{A}}(T)}{T} - \frac{k\epsilon_0}{\gamma},$$

*where $S$ is the distribution which uniformly assigns to each example $(x_i, y_i)$ probability $1/m$.*

The proof of Theorem 5 is split over the next two subsections.

## A.2 Improper game playing

An important step towards proving Theorem 5 is framing statistical agnostic multiclass boosting as an improper zero-sum game. A similar idea was used by Brukhim et al. [6] for binary classification. We take a detour and elaborate on this connection here.

Consider an improper zero-sum game. In this game, there are two players A and B, and a payoff function $g$ that decomposes into the sum of $m$ smaller independent convex-concave payoff functions $f^1, ..., f^m$ each of which depends on the players' strategies. The goal for Player A is to minimize $g$, while the goal for Player B is to maximize $g$. Let $\mathcal{K}_A$ and $\mathcal{K}_B$ be the convex, compact decisions sets of players A and B respectively. In addition, let $\mathcal{K}_C$ be a convex, compact set, and let $\mathcal{K}_A$ be a matrix consisting of $m$ row vectors from $\mathcal{K}_C$. Because convexity and concavity are preserved under

(non-negative weighted) summation, $g$ is also convex-concave. By Sion's minimax theorem [34], the value of the game is well-defined, which we denote as

$$\lambda^* = \min_{P \in \mathcal{K}_A} \max_{q \in \mathcal{K}_B} g(P, q) = \max_{q \in \mathcal{K}_B} \min_{P \in \mathcal{K}_a} g(P, q).$$

Let $\mathcal{K}'_B$ be a convex compact set such that $\mathcal{K}_B \subseteq \mathcal{K}'_B$. Strategies in $\mathcal{K}_B$ are proper, while those in $\mathcal{K}'_B$ are improper. We allow $g$ to be defined over $\mathcal{K}'_B$. In addition, we make the following three assumptions:

1. Player B has access to a randomized approximate optimization oracle $\mathcal{W}$. Given any $P \in \mathcal{K}_\mathcal{A}$, $\mathcal{W}$ outputs a response $q \in \mathcal{K}'_B$ such that $\mathbb{E}\left[g(P, q)\right] \geq \max_{q^* \in \mathcal{K}_B} g(P, q^*) - \epsilon$, where the expectation is over randomness of $\mathcal{W}$

2. Player B is allowed to play strategies in $\mathcal{K}'_B$

3. Player A has access to $m$ copies of possibly (independently) randomized OCO($\mathcal{K}_C$, $T$) algorithms $\mathcal{A}_1, ..., \mathcal{A}_m$, all with regret $R_\mathcal{A}(T)$

---

**Algorithm 3:** Improper Game Playing

---
1  **for** $t = 1, ..., T$ **do**
2  |   Player A plays $P_t$
3  |   Player B plays $q_t \in \mathcal{K}'_B$, where $q_t = \mathcal{W}(P_t)$
4  |   Player A and B lose/gain payoff $g(P_t, q_t) = \sum_{i=1}^{m} f^i(P_t[i], q_t)$
5  |   Define loss: $\ell_t^i(p) = f^i(p, q_t) \, \forall i \in [m]$
6  |   Player A updates $P_{t+1}[i] = \mathcal{A}^i(\ell_1^i, ..., \ell_t^i) \, \forall i \in [m]$

---

**Proposition 6.** *If players A and B play according to Algorithm 3, then player B's average strategy $\bar{q} = \frac{1}{T} \sum_{t=1}^{T} q_t, \bar{q} \in \mathcal{K}'_B$, satisfies for any $P^* \in \mathcal{K}_A$,*

$$\lambda^* \leq \mathbb{E}\left[g(P^*, \bar{q})\right] + \frac{m R_\mathcal{A}(T)}{T} + \epsilon,$$

*where the expectation is over the randomness of $\mathcal{W}$.*

*Proof.* Since the game is well-defined over $\mathcal{K}_A$ and $\mathcal{K}_\mathcal{B}$, there exists a max-min strategy $q^* \in \mathcal{K}_B$ for player B such that for all $P \in \mathcal{K}_A$, $g(P, q^*) \geq \lambda^*$. Let $\bar{P} = \frac{1}{T} \sum_{t=1}^{T} P_t$. Then,

$$
\begin{aligned}
\mathbb{E}\left[\frac{1}{T} \sum_{t=1}^{T} g(P_t, q_t)\right] &\geq \mathbb{E}\left[\frac{1}{T} \sum_{t=1}^{T} \max_{q \in \mathcal{K}_B} g(P_t, q)\right] - \epsilon \\
&\geq \mathbb{E}\left[\frac{1}{T} \sum_{t=1}^{T} g(P_t, q^*)\right] - \epsilon \\
&\geq \mathbb{E}\left[d(\bar{P}, q^*)\right] - \epsilon \\
&\geq \lambda^* - \epsilon.
\end{aligned}
$$

Now let $\bar{q} = \frac{1}{T} \sum_{t=1}^{T} q_t$. Note that $\bar{q} \in \mathcal{K}'_B$. Then we write:

$$\mathbb{E}\left[\frac{1}{T}\sum_{t=1}^{T}g(P_t,q_t)\right] = \mathbb{E}\left[\frac{1}{T}\sum_{t=1}^{T}\sum_{i=1}^{m}f^i(P_t[i],q_t)\right]$$

$$= \sum_{i=1}^{m}\mathbb{E}\left[\frac{1}{T}\sum_{t=1}^{T}f^i(P_t[i],q_t)\right]$$

$$\leq \sum_{i=1}^{m}\left(\mathbb{E}\left[\frac{1}{T}\sum_{t=1}^{T}f^i(P^*[i],q_t)\right] + \frac{R_{\mathcal{A}}(T)}{T}\right)$$

$$= \mathbb{E}\left[\frac{1}{T}\sum_{t=1}^{T}\sum_{i=1}^{m}f^i(P^*[i],q_t)\right] + \frac{mR_{\mathcal{A}}(T)}{T}$$

$$= \mathbb{E}\left[\frac{1}{T}\sum_{t=1}^{T}g(P^*,q_t)\right] + \frac{mR_{\mathcal{A}}(T)}{T}$$

$$\leq \mathbb{E}\left[g(P^*,\bar{q})\right] + \frac{mR_{\mathcal{A}}(T)}{T}.$$

Here, $P^*$ is any arbitrary matrix in $\mathcal{K}_A$. Combining the lower and upper bounds, we get:

$$\lambda^* \leq \mathbb{E}\left[g(P^*,\bar{q})\right] + \frac{mR_{\mathcal{A}}(T)}{T} + \epsilon,$$

which completes the proof. $\qquad\square$

### A.3 Proof of Theorem 5

Now we are ready to prove Theorem 5. The proof strategy will be as follows. We will first show how the proposed agnostic boosting algorithm is an instance of the improper game playing setup described above. Then, we will show that a weak learner satisfying Definition 2 corresponds to the randomized approximate optimization oracle. Finally, we will explicitly compute the value of the game, and derive the lower bound on correlation.

Under Definition 2, we can carefully argue that the agnostic weak learner $\mathcal{W}$ induces an approximate optimization oracle for Player B. Specifically, we can show the following lemma.

**Lemma 7.** *For any $P \in \mathcal{K}_A$, the output $q' = \frac{\mathcal{W}(P)}{\gamma} \in \mathcal{K}'_B$ satisfies,*

$$\mathbb{E}\left[g(P,q')\right] \geq \max_{q \in \mathcal{K}_B} g(P,q) - \frac{mk\epsilon_0}{\gamma}.$$

*Proof.* Note that in line 3 of Algorithm 2, we pass re-labelled examples back to the weak learner. An alternative and equivalent approach to what is presented is to first relabel each example using $P$ and then to uniformly sample and pass $m_0$ examples to $\mathcal{W}$. Let $\tilde{y}_i$ correspond to the relabeled class of example $i$. Recall that $\tilde{y}_i \sim P[i]$. Let $R$ be the distribution over $(x_i, \tilde{y}_i)$ after relabelling and define $h^* = \max_{h \in \mathcal{H}} cor_R(h)$. Then, by the weak learning assumption,

$$\mathop{\mathbb{E}}_{S'|\tilde{y}} \left[ \sum_{i=1}^{m} \mathcal{W}_{S'}(x_i) \cdot (2\tilde{y}_i - \mathbb{1}_k) \right] = \mathop{\mathbb{E}}_{S'|\tilde{y}} \left[ \sum_{\ell \in \mathcal{B}_k} \sum_{i:\tilde{y}_i = \ell} \mathcal{W}_{S'}(x_i) \cdot (2\tilde{y}_i - \mathbb{1}_k) \right]$$

$$\geq \sum_{\ell \in \mathcal{B}_k} \left( \gamma \max_{h \in \mathcal{H}} \sum_{i:\tilde{y}_i = \ell} h(x_i) \cdot (2\tilde{y}_i - \mathbb{1}_k) - m\epsilon_0 \right)$$

$$\geq \gamma \sum_{\ell \in \mathcal{B}_k} \sum_{i:\tilde{y}_i = \ell} h^*(x_i) \cdot (2\tilde{y}_i - \mathbb{1}_k) - mk\epsilon_0$$

$$= \gamma \sum_{i=1}^{m} h^*(x_i) \cdot (2\tilde{y}_i - \mathbb{1}_k) - mk\epsilon_0.$$

Taking the expectation of both sides,

$$\mathop{\mathbb{E}}_{\tilde{y}} \left[ \mathop{\mathbb{E}}_{S'|\tilde{y}} \left[ \sum_{i=1}^{m} \mathcal{W}_{S'}(x_i) \cdot (2\tilde{y}_i - \mathbb{1}_k) \right] \right] = \mathbb{E} \left[ \sum_{i=1}^{m} \mathcal{W}_{S'}(x_i) \cdot (2\tilde{y}_i - \mathbb{1}_k) \right]$$

$$= \mathop{\mathbb{E}}_{\tilde{S}'} \left[ \mathop{\mathbb{E}}_{\tilde{y}|S'} \left[ \sum_{i=1}^{m} \mathcal{W}_{S'}(x_i) \cdot (2\tilde{y}_i - \mathbb{1}_k) \right] \right]$$

$$\geq \mathop{\mathbb{E}}_{\tilde{y}} \left[ \gamma \sum_{i=1}^{m} h^*(x_i) \cdot (2\tilde{y}_i - \mathbb{1}_k) \right] - mk\epsilon_0.$$

For any $h \in \mathcal{H}$,

$$\mathop{\mathbb{E}}_{\tilde{y}} \left[ \frac{1}{m} \sum_{i=1}^{m} h(x_i) \cdot (2\tilde{y}_i - \mathbb{1}_k) \right] = \mathop{\mathbb{E}}_{(x_i, y_i) \sim R} [h(x_i) \cdot (2y_i - \mathbb{1}_k)]$$

$$= \mathop{\mathbb{E}}_{x_i} \left[ \mathop{\mathbb{E}}_{y_i|x_i} [h(x_i) \cdot (2y_i - \mathbb{1}_k)] \right]$$

$$= \mathop{\mathbb{E}}_{x_i} [h(x_i) \cdot (2 \mathbb{E}[y_i|x_i] - \mathbb{1}_k)]$$

$$= \mathop{\mathbb{E}}_{x_i} [h(x_i) \cdot (2P[i] - \mathbb{1}_k)]$$

$$= \frac{1}{m} \sum_{i=1}^{m} h(x_i) \cdot (2P[i] - \mathbb{1}_k)$$

$$= \mathop{\mathbb{E}}_{(x_i, y_i) \sim D} [h(x_i) \cdot (2P[i] - \mathbb{1}_k)].$$

Note the last equality follows from the fact that both $R$ and $D$ have the same marginal distribution over unlabeled examples $x_i$'s. Combining the results, multiplying by $m$, and dividing by $\gamma$ gives,

$$\mathop{\mathbb{E}}_{S'} \left[ \sum_{i=1}^{m} \frac{\mathcal{W}_{S'}(x_i)}{\gamma} \cdot (2P[i] - \mathbb{1}_k) \right] \geq \sum_{i=1}^{m} h^*(x_i) \cdot (2P[i] - \mathbb{1}_k) - \frac{mk\epsilon_0}{\gamma}.$$

Finally, note that $q'(x_i) = \frac{\mathcal{W}_{S'}(x_i)}{\gamma} \in \mathcal{K}'_B$. Then, recalling our definition of $g$,

$$\mathbb{E}\left[g(P, q')\right] = \mathbb{E}\left[\sum_{i=1}^{m}(2P[i] - \mathbb{1}_k) \cdot (q'(x_i) - y_i)\right]$$

$$= \mathbb{E}\left[\sum_{i=1}^{m}(2P[i] - \mathbb{1}_k) \cdot q'(x_i)\right] - \sum_{i=1}^{m}(2P[i] - \mathbb{1}_k) \cdot y_i$$

$$\geq \sum_{i=1}^{m} h^*(x_i) \cdot (2P[i] - \mathbb{1}_k) - \frac{mk\epsilon_0}{\gamma} - \sum_{i=1}^{m}(2P[i] - \mathbb{1}_k) \cdot y_i$$

$$= \sum_{i=1}^{m}(h^*(x_i) - y_i) \cdot (2P[i] - \mathbb{1}_k) - \frac{mk\epsilon_0}{\gamma}$$

$$= \max_{q \in \mathcal{K}_B} g(P, q) - \frac{mk\epsilon_0}{\gamma}.$$

which completes the proof.

$\square$

Now, we return to proving Theorem 5. We start by explicitly computing the value of the above game. One can show that the dominant strategy for player A is to return a matrix where row $i$ is the vector $y_i$. That is, $P = [y_1; y_2; ...; y_m]$. Because $g$ decomposes into the sum of smaller independent losses $f$, it is helpful to instead focus our attention on

$$f^i(P[i], q) = (2P[i] - \mathbb{1}_k) \cdot (q(x_i) - y_i).$$

Above, Player A has control over the vector $P[i]$ and wants to minimize $f^i$. We can show that regardless of what $q(x_i) \in \mathcal{K}_B$ is, Player A minimizes $f$ by playing $P[i] = y_i$. Note, only the case where $q(x_i) \neq y_i$ is important. Under this scenario, Player A needs to maximize the value at the $y_i$th index and minimize the value at the $q(x_i)$th index. This is precisely accomplished by setting $P[i] = y_i$. Since the smaller loss functions $f^i$ are independent of one another, it follows that Player A minimizes $g$ by playing matrix $P = [y_1; y_2; ...; y_m]$. Under this fixed dominant strategy for A, the value of the game $\lambda^*$ can be computed as

$$\lambda^* = \max_{q \in \mathcal{K}_B} \min_{P \in \mathcal{K}_a} g(P, q)$$

$$= \max_{q \in \mathcal{K}_B} \sum_{i=1}^{m}(2y_i - \mathbb{1}_k) \cdot (q(x_i) - y_i)$$

$$= \max_{q \in \mathcal{K}_B} \sum_{i=1}^{m} 2(q(x_i) \cdot y_i - 1)$$

$$= m \cdot cor_S(h^*) - m.$$

Then, for any $P^*$ using Proposition 6,

$$m \cdot \max_{h \in \mathcal{H}} cor_S(h) - m \leq \mathbb{E}\left[g(P^*, \bar{q})\right] + \frac{mR_{\mathcal{A}}(T)}{T} + \frac{mk\epsilon_0}{\gamma}$$

$$= \mathbb{E}\left[\sum_{i=1}^{m}(2P^*[i] - \mathbb{1}_k) \cdot (\bar{q}(x_i) - y_i)\right] + \frac{mR_{\mathcal{A}}(T)}{T} + \frac{mk\epsilon_0}{\gamma}$$

$$= \mathbb{E}\left[\sum_{i=1}^{m}(2P^*[i] - \mathbb{1}_k) \cdot (\frac{1}{\gamma T}\sum_{t=1}^{T}h_t(x_i) - y_i)\right] + \frac{mR_{\mathcal{A}}(T)}{T} + \frac{mk\epsilon_0}{\gamma}$$

$$= \mathbb{E}\left[\sum_{i=1}^{m}P^*[i] \cdot \left(\frac{\frac{2}{T}\sum_{t=1}^{T}h_t(x_i) - \mathbb{1}_k}{\gamma} - (2y_i - \mathbb{1}_k)\right)\right]$$

$$= \quad + \frac{mR_{\mathcal{A}}(T)}{T} + \frac{mk\epsilon_0}{\gamma}. \qquad \text{(Lemma 19)}$$

According to Lemma 17, there exists a $P^*[i]$ and therefore a $P^*$, such that

$$m \cdot \max_{h \in \mathcal{H}} cor_S(h) - m \leq \mathbb{E}\left[\sum_{i=1}^{m}2\left(\prod\left(\frac{\sum_{i=1}^{T}h_t(x_i)}{\gamma T}\right) \cdot y_i - 1\right)\right] + \frac{mR_{\mathcal{A}}(T)}{T} + \frac{mk\epsilon_0}{\gamma}$$

$$= \mathbb{E}\left[\sum_{i=1}^{m}2\bar{h}(x_t) \cdot y_i - 1\right] - m + \frac{mR_{\mathcal{A}}(T)}{T} + \frac{mk\epsilon_0}{\gamma}.$$

Subtracting and then dividing both sides by $m$,

$$\max_{h \in \mathcal{H}} cor_S(h) \leq \mathbb{E}\left[\frac{1}{m}\sum_{i=1}^{m}2\bar{h}(x_t) \cdot y_i - 1\right] + \frac{R_{\mathcal{A}}(T)}{T} + \frac{k\epsilon_0}{\gamma}$$

$$= \mathbb{E}\left[cor_S(\bar{h})\right] + \frac{R_{\mathcal{A}}(T)}{T} + \frac{k\epsilon_0}{\gamma}.$$

Rearranging, we have shown that

$$\mathbb{E}\left[cor_S(\bar{h})\right] \geq \max_{h \in \mathcal{H}} cor_S(h) - \frac{R_{\mathcal{A}}(T)}{T} - \frac{k\epsilon_0}{\gamma}$$

which completes the proof. If one further lets $R_{\mathcal{A}}(T)$ to be OGD, as in the online setting, then observe that we get

$$\mathbb{E}\left[cor_S(\bar{h})\right] \geq \max_{h \in \mathcal{H}} cor_S(h) - \frac{R_{\mathcal{A}}(T)}{T} - \frac{k\epsilon_0}{\gamma}$$

## B    Realizable Multiclass Boosting

In this section, we fill in the gaps of Section 4 for the online and statistical realizable settings by providing boosting algorithms and their theoretical analysis. Namely, we show explicitly how the OCO framework can also be used to construct multiclass boosting algorithms in the realizable setting. Unlike in the agnostic setting, our realizable boosting algorithms will update weak learners by *reweighting* examples.

## B.1 Online Setting

We now describe a online realizable boosting algorithm whose pseudocode is provided below. The booster has black-box oracle access to $N$ copies of a $(\gamma, T)$-RWOL algorithm $\mathcal{W}_1, ..., \mathcal{W}_N$, each satisfying Definition 3, and an OCO$([0,1], N)$ algorithm $\mathcal{A}$. Importantly, note that in line 4 , we denote by $\prod$ the $L_2$ projection onto $\Delta_k$.

---

**Algorithm 4:** Online Realizable Multiclass Boosting via OCO

**Input:** $(\gamma, T)$-RWOL $\mathcal{W}_1...\mathcal{W}_N$, OCO$([0,1], N)$ algorithm $\mathcal{A}$

1 **for** $t = 1, ..., T$ **do**
2      Receive example $x_t$
3      Accumulate weak predictions $h_t = \sum_{i=1}^N \mathcal{W}_i(x_t)$
4      Set $\mathcal{D}_t = \Pi(\frac{h_t}{\gamma N})$
5      Predict $\hat{y}_t \sim \mathcal{D}_t$
6      Receive true label $y_t$
7      **for** $i = 1, ..., N$ **do**
8          If $i > 1$, obtain $p_t^i = \mathcal{A}(l_t^1, ..., l_t^{i-1})$. Else, initialize $p_t^1 = 0.5$.
9          Reveal loss function: $l_t^i(p) = p(\frac{2\mathcal{W}_i(x_t) \cdot y_t - 1}{\gamma} - 1)$
10          Pass $(x_t, y_t)$ to $\mathcal{W}_i$ w.p. $p_t^i$
11      Reset $\mathcal{A}$

---

**Theorem 8.** *The regret bound of Algorithm 4 satisfies:*

$$\frac{1}{T} \sum_{t=1}^T (2\,\mathbb{E}\,[\hat{y}_t] \cdot y_t - 1) \geq 1 - \frac{R_{\mathcal{A}}(N)}{N} - \frac{\tilde{R}_{\mathcal{W}}(T, k)}{\gamma T},$$

*where* $\tilde{R}_{\mathcal{W}}(T, k) = 2R_{\mathcal{W}}(T, k) + \tilde{O}(\sqrt{T}).$

*Proof.* As usual, the approach is to compute a lower and upper bound on the sum of the expected losses passed to the OCO oracle. Starting with the upper bound,

$$
\begin{aligned}
\mathbb{E}\left[\frac{1}{N} \sum_{i=1}^N l_t^i(p_t^i)\right] &\leq \frac{1}{N} \mathbb{E}\left[\min_{p \in \mathcal{K}} \sum_{i=1}^N l_t^i(p)\right] + \frac{R_{\mathcal{A}}(N)}{N} \quad &\text{(OCO Regret)} \\
&\leq \frac{1}{N} \mathbb{E}\left[\sum_{i=1}^N l_t^i(p^*)\right] + \frac{R_{\mathcal{A}}(N)}{N} \\
&= \frac{1}{N} \sum_{i=1}^N \mathbb{E}\left[p^* \left(\frac{2\mathcal{W}_i(x_t) \cdot y_t - 1}{\gamma} - 1\right)\right] + \frac{R_{\mathcal{A}}(N)}{N} \\
&= \mathbb{E}\left[p^* \left(\frac{2\left(\frac{1}{N}\sum_{i=1}^N \mathcal{W}_i(x_t)\right) \cdot y_t - 1}{\gamma} - 1)\right)\right] + \frac{R_{\mathcal{A}}(N)}{N} \\
&\leq \mathbb{E}\left[2\left(\prod\left(\frac{1}{\gamma N} \sum_{i=1}^N \mathcal{W}_i(x_t)\right) \cdot y_t - 1\right)\right] + \frac{R_{\mathcal{A}}(N)}{N} \quad &\text{(Lemma 18)} \\
&= 2\left(\mathbb{E}\,[\hat{y}_t] \cdot y_t - 1\right) + \frac{R_{\mathcal{A}}(N)}{N}.
\end{aligned}
$$

Finally, summing over $t \in [T]$,

$$\mathbb{E}\left[\frac{1}{N}\sum_{t=1}^{T}\sum_{i=1}^{N}l_t^i(p_t^i)\right] \leq \sum_{t=1}^{T}2\left(\mathbb{E}\left[\hat{y}_t\right]\cdot y_t - 1\right) + \frac{TR_{\mathcal{A}}(N)}{N}.$$

For the lower bound, we first need the following important Lemma adapted from [4] to the multiclass setting:

**Lemma 9.** *For any weak learner $(\gamma, T)$-RWOL $\mathcal{W}$, there exists $\tilde{R}_{\mathcal{W}}(T,k) = \tilde{\mathcal{O}}(\sqrt{\sum_t p_t}) + 2R_{\mathcal{W}}(T,k)$ such that for any sequence $p_1, ..., p_T \in [0,1]$,*

$$\sum_{t=1}^{T}p_t(2\mathcal{W}(x_t)\cdot y_t - 1) \geq \gamma\sum_{t=1}^{T}p_t - \tilde{R}_{\mathcal{W}}(T,k).$$

The proof of Lemma 9 follows exactly from Lemma 14 in Brukhim et al. [6] and Lemma 1 in Beygelzimer et al. [4] and so we omit it here. Using Lemma 9,

$$\frac{1}{\gamma}\mathbb{E}\left[\sum_{i=1}^{N}\sum_{t=1}^{T}p_t^i(2\mathcal{W}(x_t)\cdot y_t - 1)\right] \geq \mathbb{E}\left[\frac{1}{\gamma}\sum_{i=1}^{N}\left(\gamma\sum_{t=1}^{T}p_t^i - \tilde{R}_{\mathcal{W}}(T,k)\right)\right]$$
$$= \sum_{i=1}^{N}\sum_{t=1}^{T}\mathbb{E}\left[p_t^i\right] - \frac{N}{\gamma}\tilde{R}_{\mathcal{W}}(T,k).$$

Then,

$$\mathbb{E}\left[\sum_{t=1}^{T}\sum_{i=1}^{N}l_t^i(p_t^i)\right] = \frac{1}{\gamma}\mathbb{E}\left[\sum_{i=1}^{N}\sum_{t=1}^{T}p_t^i(2\mathcal{W}(x_t)\cdot y_t - 1)\right] - \sum_{i=1}^{N}\sum_{t=1}^{T}\mathbb{E}\left[p_t^i\right] \geq -\frac{N}{\gamma}\tilde{R}_{\mathcal{W}}(T,k).$$

By combining upper and lower bounds for $\mathbb{E}\left[\frac{1}{NT}\sum_t\sum_i l_t^i(p_t^i)\right]$, we get

$$\frac{1}{T}\sum_{t=1}^{T}(2\,\mathbb{E}\left[\hat{y}_t\right]\cdot y_t - 1) \geq 1 - \frac{R_{\mathcal{A}}(N)}{N} - \frac{\tilde{R}_{\mathcal{W}}(T,k)}{\gamma T}.$$

which completes the proof. $\qquad\square$

## B.2 Statistical Setting

We now describe a statistical realizable boosting algorithm whose pseudocode is provided below. The booster is given as input a sample $S = (x_1, y_1), ..., (x_m, y_m) \in \mathcal{X} \times \mathcal{B}_k$. The booster has black-box oracle access to $T$ copies of a $(\gamma, m_0)$-RWL algorithm $\mathcal{W}_1, ..., \mathcal{W}_T$, each satisfying Definition 4, and $m$ copies of an OCO$([0,1], T)$ algorithm, $\mathcal{A}^1, ..., \mathcal{A}^m$. Importantly, note that in line 8, we denote by $\prod$ as a randomized *prediction* operator that given a $\gamma$-scaled distribution over $k$ labels, first computes the $L_2$ projection onto $\Delta_k$, and then randomly samples a label from the resulting distribution.

**Algorithm 5:** Statistical Realizable Boosting via OCO

**Input:** $(\gamma, m_0)$-RWLs $\mathcal{W}_1...\mathcal{W}_T$, OCO$([0,1], T)$ algorithms $\mathcal{A}^1...\mathcal{A}^m$, and Sample
$\qquad S = (x_1, y_1)...(x_m, y_m) \in \mathcal{X} \times \mathcal{B}_k$

1 **for** $t = 1, ..., T$ **do**
2 $\quad$ If $t = 1$, set $P_1[i] = 1/m \ \forall i \in [m]$
3 $\quad$ Pass $m_0$ examples to $\mathcal{W}_t$ drawn from the following distribution:
4 $\quad$ Draw $(x_i, y_i)$ w.p. $\propto P_t[i]$
5 $\quad$ Let $h_t$ be the weak hypothesis returned by $\mathcal{W}_t$
6 $\quad$ Reveal loss function: $l_t^i(p) = p(\frac{h_t(x_i)}{\gamma} \cdot (2y_i - \mathbb{1}_k) - 1) \ \forall i \in [m]$
7 $\quad$ Set $P_{t+1}[i] = \mathcal{A}^i(l_1^i, ..., l_t^i) \ \forall i \in [m]$
8 return $\bar{h}(x) = \prod \left( \frac{1}{\gamma T} \sum_{t=1}^T h_t(x) \right)$

**Theorem 10.** *The output of Algorithm 5, which is denoted $\bar{h}$, satisfies,*

$$\mathbb{E}\left[cor_S(\bar{h})\right] \geq 1 - \frac{R_{\mathcal{A}}(T)}{T},$$

*where $S$ is the distribution which uniformly assigns to each example $(x_i, y_i)$ probability $1/m$.*

*Proof.* We will use a very similar proof strategy as in the agnostic setting by reducing to improper game playing. Let $h^*$ be a hypothesis consistent with the input sample (i.e. $h^*(x_i) = y_i$ for $i \leq m$) and let $\mathcal{H}' = \mathcal{H} \cup h^*$. We begin by establishing the reduction to improper game playing. Take $\mathcal{K}_A = [0, 1]^m$, $\mathcal{K}_B = \Delta_{\mathcal{H}}$, $\mathcal{K}_B' = \frac{1}{\gamma}\mathcal{K}_B$. Define payoff functions:

$$f^i(p, q) = p(q(x_i) \cdot (2y_i - \mathbb{1}_k) - 1)$$

$$\begin{aligned} g(P, q) &= \sum_{i=1}^m f^i(P[i], q) \\ &= \sum_{i=1}^m P[i](q(x_i) \cdot (2y_i - \mathbb{1}_k) - 1). \end{aligned}$$

We will now show that the agnostic weak learner $\mathcal{W}$ induces an approximate optimization oracle for Player B. Specifically, we will show the following lemma.

**Lemma 11.** *For any $P \in \mathcal{K}_A$, the output $q' = \frac{\mathcal{W}(P)}{\gamma} \in \mathcal{K}_B'$ satisfies*

$$\mathbb{E}\left[g(P, q')\right] \geq 0.$$

*That is, the weak learner corresponds to an approximate oracle with $0$ error.*

*Proof.* Using the definition of correlation and the weak learning condition.

$$\mathbb{E}\left[cor_P(\mathcal{W}_{\mathcal{S}'})\right] = \mathbb{E}\left[\sum_{i=1}^m \frac{P[i]}{\sum_{i=1}^m P[i]} \mathcal{W}_{\mathcal{S}'}(x_i) \cdot (2y_i - \mathbb{1}_k)\right] \geq \gamma$$

and therefore,

$$\mathbb{E}\left[\sum_{i=1}^m P[i] \frac{\mathcal{W}_{\mathcal{S}'}(x_i)}{\gamma} \cdot (2y_i - \mathbb{1}_k)\right] \geq \sum_{i=1}^m P[i].$$

Now, take $q'(x_i) = \frac{\mathcal{W}_{\mathcal{S}'}(x_i)}{\gamma} \in \mathcal{K}_B'$. Then, recalling our definition of $g$,

$$\mathbb{E}\left[g(P, q')\right] = \mathbb{E}\left[\sum_{i=1}^{m} P[i](q'(x_i) \cdot (2y_i - \mathbb{1}_k) - 1)\right]$$

$$= \mathbb{E}\left[\sum_{i=1}^{m} P[i]q'(x_i) \cdot (2y_i - \mathbb{1}_k)\right] - \sum_{i=1}^{m} P[i]$$

$$\geq 0 = \max_{q \in \mathcal{K}_B} g(P, q).$$

$\square$

Now, we will complete the last part of the overall proof. We start by explicitly computing the value of the above game. One can show that the dominant strategy for player B is to return $h^*$. Indeed, since $h^*$ is consistent, we can show that $g(P, h^*) = 0$ for any $P$ and thus $\lambda^* = 0$. Then, for any $P^*$ using Proposition 6,

$$0 \leq \mathbb{E}\left[g(P^*, \bar{q})\right] + \frac{mR_{\mathcal{A}}(T)}{T}$$

$$= \mathbb{E}\left[\sum_{i=1}^{m} P^*[i](\bar{q} \cdot (2y_i - \mathbb{1}_k) - 1)\right] + \frac{mR_{\mathcal{A}}(T)}{T}$$

$$= \mathbb{E}\left[\sum_{i=1}^{m} P^*[i]\left(\left(\frac{1}{\gamma T}\sum_{t=1}^{T} h_t(x_i)\right) \cdot (2y_i - \mathbb{1}_k) - 1\right)\right] + \frac{mR_{\mathcal{A}}(T)}{T}$$

$$= \mathbb{E}\left[\sum_{i=1}^{m} P^*[i]\left(\frac{\left(\frac{2}{T}\sum_{t=1}^{T} h_t(x_i)\right) \cdot y_i - 1}{\gamma} - 1\right)\right] + \frac{mR_{\mathcal{A}}(T)}{T}.$$

According to Lemma 18, there exists a $P^*[i]$ and therefore a $P^*$, such that

$$0 \leq \mathbb{E}\left[\sum_{i=1}^{m} 2\left(\prod\left(\frac{\sum_{i=1}^{T} h_t(x_i)}{\gamma T}\right) \cdot y_i - 1\right)\right] + \frac{mR_{\mathcal{A}}(T)}{T}$$

$$= \mathbb{E}\left[\sum_{i=1}^{m} 2\bar{h}(x_t) \cdot y_i - 1\right] - m + \frac{mR_{\mathcal{A}}(T)}{T}.$$

Dividing both sides by $m$,

$$0 \leq \mathbb{E}\left[\frac{1}{m}\sum_{i=1}^{m} 2\bar{h}(x_t) \cdot y_i - 1\right] - 1 + \frac{R_{\mathcal{A}}(T)}{T}$$

$$= \mathbb{E}\left[cor_S(\bar{h})\right] - 1 + \frac{R_{\mathcal{A}}(T)}{T}.$$

Thus, rearranging, we have shown that

$$\mathbb{E}\left[cor_S(\bar{h})\right] \geq 1 - \frac{R_{\mathcal{A}}(T)}{T}.$$

$\square$

# C Alternate Weak Learning Conditions

Definition 1 roughly requires that the online weak learner be able to distinguish between every pair of classes to some non-trivial, but far from optimal, degree. In this section, we propose other possible online agnostic weak learning conditions *without* changing Algorithm 1.

We start by first considering a setting where there always exists a hypothesis $h \in \mathcal{H}$ that, in expectation, makes at most $T/2$ mistakes (i.e. gets at least $1/2$ correct). Here, the weak learning condition below suffices as $\max_{h \in \mathcal{H}} \mathbb{E} \left[ \sum_{t=1}^{T} 2h(x_t) \cdot y_t - 1 \right]$ is guaranteed to be positive.

**Definition 5.** *Let $\mathcal{H} \subseteq \mathcal{B}_k^{\mathcal{X}}$ be a class of experts and let $0 < \gamma \leq 1$ denote the "advantage". An online learning algorithm $\mathcal{W}$ is a $(\gamma, T)$-agnostic weak online learner for $\mathcal{H}$ if for any adaptively chosen sequence of tuples $(x_t, y_t) \in \mathcal{X} \times \mathcal{B}_k$, at every iteration $t \in [T]$, the algorithm outputs $\mathcal{W}(x_t) \in \mathcal{B}_k$ such that,*

$$\gamma \max_{h \in \mathcal{H}} \mathbb{E} \left[ \sum_{t=1}^{T} 2h(x_t) \cdot y_t - 1 \right] - \mathbb{E} \left[ \sum_{t=1}^{T} 2\mathcal{W}(x_t) \cdot y_t - 1 \right] \leq R_{\mathcal{W}}(T, k),$$

*where the expectation is taken w.r.t. the randomness of the weak learner $\mathcal{W}$ and that of the possibly adaptive adversary, $R_{\mathcal{W}} : \mathbb{N} \times \mathbb{N} \to \mathbb{R}_+$ is the additive regret: a non-decreasing, sub-linear function of $T$.*

For weak learners satisfying Definition 5, one can show, using the same proof strategy in Section 3.2, that Algorithm 1 achieves the average regret bound below.

**Proposition 12.** *Assuming weak learners satisfy Definition 5, the expected regret bound of Algorithm 1 is*

$$\frac{1}{T} \max_{h \in \mathcal{H}} \mathbb{E} \left[ \sum_{t=1}^{T} 2h(x_t) \cdot y_t - 1 \right] - \frac{1}{T} \mathbb{E} \left[ \sum_{t=1}^{T} 2\hat{y}_t \cdot y_t - 1 \right] \leq \frac{R_{\mathcal{W}}(T, k)}{\gamma T} + \frac{R_{\mathcal{A}}(N)}{N}.$$

For completeness sake, we include a partial proof below.

*Proof.* The proof follows the same strategy as that used to prove Theorem 1. That is, we will upper and lower bound the expected sum of loss passed to the OCO algorithm. Fortunately, the proof of the upper bound remains exactly the same as that in the proof of Theorem 1. Thus, we will only derive a lower bound on the sum of losses passed to the OCO algorithm. Define $h^* = \arg \max_{h \in \mathcal{H}} \sum_{t=1}^{T} (2h(x_t) \cdot y_t - 1)$ as the optimal competitor in hindsight. Then,

$$\mathbb{E} \left[ \sum_{i=1}^{N} \sum_{t=1}^{T} l_t^i(p_t^i) \right] = \mathbb{E} \left[ \sum_{i=1}^{N} \sum_{t=1}^{T} p_t^i \cdot \left( \frac{2\mathcal{W}_i(x_t) - \mathbb{1}_k}{\gamma} - (2y_t - \mathbb{1}_k) \right) \right]$$

$$= \frac{1}{\gamma} \sum_{i=1}^{N} \sum_{t=1}^{T} \mathbb{E} \left[ p_t^i \cdot (2\mathcal{W}_i(x_t) - \mathbb{1}_k) \right] - \sum_{i=1}^{N} \sum_{t=1}^{T} \mathbb{E} \left[ p_t^i \cdot (2y_t - \mathbb{1}_k) \right]$$

$$= \frac{1}{\gamma} \sum_{i=1}^{N} \sum_{t=1}^{T} \mathbb{E} \left[ 2\mathcal{W}_i(x_t) \cdot y_t^i - 1 \right] - \sum_{i=1}^{N} \sum_{t=1}^{T} \mathbb{E} \left[ p_t^i \cdot (2y_t - \mathbb{1}_k) \right] \quad \text{(Lemma 15)}$$

Using the weak learning condition in Definition 5,

$$\frac{1}{\gamma} \sum_{i=1}^{N} \sum_{t=1}^{T} \mathbb{E}\left[2\mathcal{W}_i(x_t) \cdot y_t^i - 1\right] \geq \sum_{i=1}^{N} \max_{h \in \mathcal{H}} \sum_{t=1}^{T} \mathbb{E}\left[2h(x_t) \cdot y_t^i - 1\right] - \frac{NR_{\mathcal{W}}(T,k)}{\gamma} \quad \text{(Definition 5)}$$

$$\geq \sum_{i=1}^{N} \sum_{t=1}^{T} \mathbb{E}\left[2h^*(x_t) \cdot y_t^i - 1)\right] - \frac{NR_{\mathcal{W}}(T,k)}{\gamma}$$

$$= \sum_{i=1}^{N} \sum_{t=1}^{T} \mathbb{E}\left[p_t^i \cdot (2h^*(x_t) - \mathbb{1}_k)\right] - \frac{NR_{\mathcal{W}}(T,k)}{\gamma}. \quad \text{(Lemma 15)}$$

Putting things together, we find,

$$\mathbb{E}\left[\sum_{i=1}^{N} \sum_{t=1}^{T} l_t^i(p_t^i)\right] \geq \sum_{i=1}^{N} \sum_{t=1}^{T} \mathbb{E}\left[p_t^i \cdot (2h^*(x_t) - 2y_t)\right] - \frac{NR_{\mathcal{W}}(T,k)}{\gamma}$$

$$\geq \sum_{i=1}^{N} \sum_{t=1}^{T} \mathbb{E}\left[2(h^*(x_t) \cdot y_t - 1)\right] - \frac{NR_{\mathcal{W}}(T,k)}{\gamma} \quad \text{(Lemma 16)}$$

$$= N \sum_{t=1}^{T} 2(h^*(x_t) \cdot y_t - 1) - \frac{NR_{\mathcal{W}}(T,k)}{\gamma}.$$

Note that this is the same lower bound as in the proof of Theorem 1 and therefore the same regret bound as in Theorem 1 holds. □

The assumption on the hypothesis class that there exists a hypothesis $h \in \mathcal{H}$ that, in expectation, makes at most $T/2$ mistakes is quite strong, especially if we allow a random adaptive adversary. Note that Definition 5 cannot be used when this assumption on $\mathcal{H}$ does not hold - $\mathbb{E}\left[\sum_{t=1}^{T} 2h(x_t) \cdot y_t - 1\right]$ can be potentially negative for every $h \in \mathcal{H}$, even for the randomly guessing competitor. One way to relax this assumption on $\mathcal{H}$ is via Definition 1 presented in the main text. Another way, is by taking a *one-vs-all* perspective to multiclass classification, which we present below.

**Definition 6.** *Let $\mathcal{H} \subseteq \mathcal{B}_k^{\mathcal{X}}$ be a class of experts and let $0 < \gamma \leq 1$ denote the "advantage". An online learning algorithm $\mathcal{W}$ is a $(\gamma, T)$-agnostic weak online learner for $\mathcal{H}$ if for any adaptively chosen sequence of tuples $(x_t, y_t) \in \mathcal{X} \times \mathcal{B}_k$, at every iteration $t \in [T]$, for every label $\ell \in \mathcal{B}_k$, the algorithm outputs $\mathcal{W}(x_t) \in \mathcal{B}_k$ such that,*

$$\gamma \max_{h \in \mathcal{H}} \mathbb{E}\left[\sum_{t:y_t=\ell} 2h(x_t) \cdot y_t - 1\right] - \mathbb{E}\left[\sum_{t:y_t=\ell} 2\mathcal{W}(x_t) \cdot y_t - 1\right] \leq R_{\mathcal{W}}(T,k),$$

*where the expectation is taken w.r.t. the randomness of the weak learner $\mathcal{W}$ and that of the possibly adaptive adversary, $R_{\mathcal{W}} : \mathbb{N} \times \mathbb{N} \to \mathbb{R}_+$ is the additive regret: a non-decreasing, sub-linear function of $T$.*

Intuitively, Definition 6 roughly requires that a weak learner be able to learn each class to a non-trivial (but far from optimal) degree of accuracy as it processes the stream of examples. In order to ensure that $\gamma \in (0, 1)$, we also need that our hypothesis class $\mathcal{H}$ contains $k$ hypothesis each of which predicts a reference class $\ell \in \mathcal{B}_k$ with probability a $1/2$. This assumption on the hypothesis class is mild: if our class $\mathcal{H}$ does not contain these randomly guessing binary hypotheses, we can add them without substantially increasing the complexity of the class $\mathcal{H}$. That said, this is still a minor drawback compared to Definition 1, where no additional assumptions on $\mathcal{H}$ are needed.

Under Definition 6, we can show that the following regret bound for Algorithm 1. Note that this regret bound is worse compared to the one derived assuming the weak learners satisfy Definition 1. Indeed, compared to Theorem 1, there is an extra factor of $k$ in the term containing $R_{\mathcal{W}(T,k)}$ in the bound below.

**Proposition 13.** *Assuming weak learners satisfy Definition 6, the expected regret bound of Algorithm 1 is*

$$\frac{1}{T} \max_{h \in \mathcal{H}} \mathbb{E} \left[ \sum_{t=1}^{T} 2h(x_t) \cdot y_t - 1 \right] - \frac{1}{T} \mathbb{E} \left[ \sum_{t=1}^{T} 2\hat{y}_t \cdot y_t - 1 \right] \leq \frac{k R_{\mathcal{W}}(T, k)}{\gamma T} + \frac{R_{\mathcal{A}}(N)}{N}.$$

*Proof.* The proof follows the same strategy as that used to prove Theorem 1. Like above, the proof of the upper bound remains exactly the same as that in the proof of Theorem 1. Thus, we will only derive a lower bound on the sum of losses passed to the OCO algorithm. Let $h^* = \arg\max_{h \in \mathcal{H}} \mathbb{E} \left[ \sum_{t=1}^{T} 2h(x_t) \cdot y_t - 1 \right]$ and $h_{i\ell}^* = \arg\max_{h \in \mathcal{H}} \mathbb{E} \left[ \sum_{t:y_t^i = \ell} 2h(x_t) \cdot y_t^i - 1 \right]$. Then,

$$\mathbb{E} \left[ \sum_{i=1}^{N} \sum_{t=1}^{T} l_t^i(p_t^i) \right] = \mathbb{E} \left[ \sum_{i=1}^{N} \sum_{t=1}^{T} p_t^i \cdot \left( \frac{2\mathcal{W}_i(x_t) - \mathbb{1}_k}{\gamma} - (2y_t - \mathbb{1}_k) \right) \right]$$

$$= \frac{1}{\gamma} \sum_{i=1}^{N} \sum_{t=1}^{T} \mathbb{E} \left[ p_t^i \cdot 2(\mathcal{W}_i(x_t) - \mathbb{1}_k) \right] - \sum_{i=1}^{N} \sum_{t=1}^{T} \mathbb{E} \left[ p_t^i \cdot (2y_t - \mathbb{1}_k) \right]$$

$$= \frac{1}{\gamma} \sum_{i=1}^{N} \sum_{t=1}^{T} \mathbb{E} \left[ 2\mathcal{W}_i(x_t) \cdot y_t^i - 1) \right] - \sum_{i=1}^{N} \sum_{t=1}^{T} \mathbb{E} \left[ p_t^i \cdot (2y_t - \mathbb{1}_k) \right] \qquad \text{(Lemma 15)}$$

$$= \frac{1}{\gamma} \sum_{i=1}^{N} \sum_{\ell \in \mathcal{B}_k} \sum_{t:y_t = \ell} \mathbb{E} \left[ 2\mathcal{W}_i(x_t) \cdot y_t^i - 1) \right] - \sum_{i=1}^{N} \sum_{t=1}^{T} \mathbb{E} \left[ p_t^i \cdot (2y_t - \mathbb{1}_k) \right]$$

Using the weak learning condition in Definition 6,

$$\frac{1}{\gamma} \sum_{i=1}^{N} \sum_{\ell \in \mathcal{B}_k} \sum_{t:y_t = \ell} \mathbb{E} \left[ 2\mathcal{W}_i(x_t) \cdot y_t^i - 1) \right] \geq \sum_{i=1}^{N} \sum_{\ell \in \mathcal{B}_k} \sum_{t:y_t = \ell} \mathbb{E} \left[ 2h_{i\ell}^*(x_t) \cdot y_t^i - 1) \right] - \frac{k N R_{\mathcal{W}}(T)}{\gamma} \quad \text{(Definition 5)}$$

$$\geq \sum_{i=1}^{N} \sum_{\ell \in \mathcal{B}_k} \sum_{t:y_t = \ell} \mathbb{E} \left[ 2h^*(x_t) \cdot y_t^i - 1) \right] - \frac{k N R_{\mathcal{W}}(T)}{\gamma}$$

$$= \sum_{i=1}^{N} \sum_{t=1}^{T} \mathbb{E} \left[ 2h^*(x_t) \cdot y_t^i - 1) \right] - \frac{k N R_{\mathcal{W}}(T)}{\gamma}$$

$$= \sum_{i=1}^{N} \sum_{t=1}^{T} \mathbb{E} \left[ p_t^i \cdot (2h^*(x_t) - \mathbb{1}_k) \right] - \frac{k N R_{\mathcal{W}}(T)}{\gamma} \qquad \text{(Lemma 15)}$$

Putting things together, we find,

$$\mathbb{E} \left[ \sum_{i=1}^{N} \sum_{t=1}^{T} l_t^i(p_t^i) \right] \geq \sum_{i=1}^{N} \sum_{t=1}^{T} \mathbb{E} \left[ p_t^i \cdot (2h^*(x_t) - 2y_t) \right] - \frac{k N R_{\mathcal{W}}(T, k)}{\gamma}$$

$$\geq \sum_{i=1}^{N} \sum_{t=1}^{T} \mathbb{E} \left[ 2(h^*(x_t) \cdot y_t - 1) \right] - \frac{k N R_{\mathcal{W}}(T, k)}{\gamma} \qquad \text{(Lemma 16)}$$

$$= N \sum_{t=1}^{T} 2(h^*(x_t) \cdot y_t - 1) - \frac{k N R_{\mathcal{W}}(T, k)}{\gamma}.$$

Combining this lower bound with the upper bound on the expected sum of losses in the proof of Theorem 1 completes this proof. □

We end this section by presenting one more online agnostic weak learning condition that places an *asymmetric* gain function on the best competitor and the weak learner. One of the benefits of this condition is that as $k$ increases, it is more explicit how exactly the weak learning assumption becomes stronger.

**Definition 7.** *Let $\mathcal{H} \subseteq \mathcal{B}_k^{\mathcal{X}}$ be a class of experts and let $0 < \gamma \leq 1$ denote the "advantage". An online learning algorithm $\mathcal{W}$ is a $(\gamma, T)$-agnostic weak online learner for $\mathcal{H}$ if for any adaptively chosen sequence of tuples $(x_t, y_t) \in \mathcal{X} \times \mathcal{B}_k$, at every iteration $t \in [T]$, the algorithm outputs $\mathcal{W}(x_t) \in \mathcal{B}_k$ such that,*

$$\gamma \max_{h \in \mathcal{H}} \mathbb{E}\left[\sum_{t=1}^{T} \frac{k h(x_t) \cdot y_t - 1}{k - 1}\right] - \mathbb{E}\left[\sum_{t=1}^{T} 2\mathcal{W}(x_t) \cdot y_t - 1\right] \leq R_{\mathcal{W}}(T, k),$$

*where the expectation is taken w.r.t. the randomness of the weak learner $\mathcal{W}$ and that of the possibly adaptive adversary, $R_{\mathcal{W}} : \mathbb{N} \times \mathbb{N} \to \mathbb{R}_+$ is the additive regret: a non-decreasing, sub-linear function of $T$.*

Along the same lines as above, under Definition 7, one can show that Algorithm 1 achieves the following average regret bound.

**Proposition 14.** *Assuming weak learners satisfy Definition 7, the expected regret bound of Algorithm 1 is*

$$\frac{1}{T} \max_{h \in \mathcal{H}} \mathbb{E}\left[\sum_{t=1}^{T} 2h(x_t) \cdot y_t - 1\right] - \frac{1}{T} \mathbb{E}\left[\sum_{t=1}^{T} 2\hat{y}_t \cdot y_t - 1\right] \leq \frac{R_{\mathcal{W}}(T, k)}{\gamma T} + \frac{R_{\mathcal{A}}(N)}{N}.$$

*Proof.* Again, we only need to show a lower bound on the expected sum of losses passed to the OCO algorithm. Define $h^* = \arg\max_{h \in \mathcal{H}} \sum_{t=1}^{T} (2h(x_t) \cdot y_t - 1)$ as the optimal competitor in hindsight. Then,

$$\mathbb{E}\left[\sum_{i=1}^{N} \sum_{t=1}^{T} l_t^i(p_t^i)\right] = \mathbb{E}\left[\sum_{i=1}^{N} \sum_{t=1}^{T} p_t^i \cdot \left(\frac{2\mathcal{W}_i(x_t) - \mathbb{1}_k}{\gamma} - (2y_t - \mathbb{1}_k)\right)\right]$$

$$= \frac{1}{\gamma} \sum_{i=1}^{N} \sum_{t=1}^{T} \mathbb{E}\left[p_t^i \cdot (2\mathcal{W}_i(x_t) - \mathbb{1}_k)\right] - \sum_{i=1}^{N} \sum_{t=1}^{T} \mathbb{E}\left[p_t^i \cdot (2y_t - \mathbb{1}_k)\right]$$

$$= \frac{1}{\gamma} \sum_{i=1}^{N} \sum_{t=1}^{T} \mathbb{E}\left[2\mathcal{W}_i(x_t) \cdot y_t^i - 1\right] - \sum_{i=1}^{N} \sum_{t=1}^{T} \mathbb{E}\left[p_t^i \cdot (2y_t - \mathbb{1}_k)\right] \qquad \text{(Lemma 15)}$$

Using the weak learning condition in Definition 7,

$$\frac{1}{\gamma} \sum_{i=1}^{N} \sum_{t=1}^{T} \mathbb{E}\left[2\mathcal{W}_i(x_t) \cdot y_t^i - 1\right] \geq \sum_{i=1}^{N} \max_{h \in \mathcal{H}} \sum_{t=1}^{T} \mathbb{E}\left[\frac{k h(x_t) \cdot y_t^i - 1}{k - 1}\right] - \frac{N R_{\mathcal{W}}(T, k)}{\gamma} \qquad \text{(Definition 7)}$$

$$\geq \sum_{i=1}^{N} \sum_{t=1}^{T} \mathbb{E}\left[2h^*(x_t) \cdot y_t^i - 1)\right] - \frac{N R_{\mathcal{W}}(T, k)}{\gamma}$$

$$= \sum_{i=1}^{N} \sum_{t=1}^{T} \mathbb{E}\left[p_t^i \cdot (2h^*(x_t) - \mathbb{1}_k)\right] - \frac{N R_{\mathcal{W}}(T, k)}{\gamma}. \qquad \text{(Lemma 15)}$$

Putting things together, we find,

$$
\begin{aligned}
\mathbb{E}\left[\sum_{i=1}^{N}\sum_{t=1}^{T} l_t^i(p_t^i)\right] &\geq \sum_{i=1}^{N}\sum_{t=1}^{T} \mathbb{E}\left[p_t^i \cdot (2h^*(x_t) - 2y_t)\right] - \frac{N R_{\mathcal{W}}(T,k)}{\gamma} \\
&\geq \sum_{i=1}^{N}\sum_{t=1}^{T} \mathbb{E}\left[2(h^*(x_t) \cdot y_t - 1)\right] - \frac{N R_{\mathcal{W}}(T,k)}{\gamma} \qquad \text{(Lemma 16)} \\
&= N \sum_{t=1}^{T} 2(h^*(x_t) \cdot y_t - 1) - \frac{N R_{\mathcal{W}}(T,k)}{\gamma}.
\end{aligned}
$$

Note that this is the same lower bound as in the proof of Theorem 1 and therefore the same regret bound as in Theorem 1 holds. $\qquad\square$

# D   Dependence of $R_{\mathcal{W}}(T,k)$ on $k$

We show concretely how the weak learner's regret, $R_{\mathcal{W}}(T,k)$, can depend on the number of classes $k$. Recall, that the weak learning condition in Definition 1 is written in terms of approximately *maximizing* a sequence of *gain* functions:

$$
\sigma_{y,\ell}(z) = \begin{cases} -1, & \text{if } z = \ell \\ 1, & \text{if } z = y \\ 0, & \text{otherwise} \end{cases}
$$

with an advantage parameter $0 < \gamma < 1$. By taking an affine transformation of $\sigma_{y,\ell}(z)$, Definition 1 can be equivalently expressed as approximately *minimizing* a sequence of bounded, non-negative *loss* functions

$$
L_{y,l}(z) = \frac{1 - \sigma_{y,\ell}(z)}{2} = \begin{cases} 1, & \text{if } z = \ell \\ 0, & \text{if } z = y \\ 1/2, & \text{otherwise} \end{cases}
$$

with an advantage parameter $\gamma > 1$. By redefining weak learning in terms of bounded, non-negative loss functions, we can tap into the rich literature of Prediction with Expert Advice to construct online agnostic weak learners. Specifically, we will use the celebrated (Randomized) Exponential Weights Algorithm (EWA). A nice fact about the EWA is that for 0-1 losses $\tilde{L}_{y_t}(z_t) = \mathbb{1}\{z_t \neq y_t\}$, and any *finite* set of experts $\mathcal{H}$, it enjoys the regret bound

$$
\mathbb{E}\left[\sum_{t=1}^{T} \tilde{L}_{y_t}(\hat{y}_t)\right] \leq \frac{\eta}{1 - e^{-\eta}} \mathbb{E}\left[\min_{h \in \mathcal{H}} \sum_{t=1}^{T} \tilde{L}_{y_t}(h(x_t))\right] + \frac{\ln(|\mathcal{H}|)}{1 - e^{-\eta}},
$$

where $\hat{y}_t$ denotes the prediction of the EWA in the $t$'th iteration and $\eta > 0$ is a tuneable learning rate (see the the exponentially-weighted forecaster from Chapter 2 of [8]). We will be using this regret bound extensively in the next two subsections.

## D.1   Finite Hypothesis Classes

In this section, we will consider two types of finite hypothesis classes: the set of discretized weight matrices and the set of discretized multiclass decision trees of depth 1.

Beginning with weight matrices, consider the multiclass classification setup with example label pairs $(x,y) \in \mathbb{R}^d \times \mathcal{B}_k$. Let $\mathcal{H}$ denote a finite hypothesis class parameterized by discretized weight matrices $W_h \in \mathbb{R}^{k \times d}$, such that $|\mathcal{H}| = \left(\frac{1}{\delta}\right)^{kd}$, where $\delta \in (0,1)$ is the level of discretization. For an input example $x \in \mathbb{R}^d$, a hypothesis $h \in \mathcal{H}$ makes its prediction as $\hat{y}_t = \arg\max_k W_h x$. Take $\mathcal{H}$ to

be a set of experts. We will now show that an instance of the EWA, using $\mathcal{H}$ and learning rate $\eta$, run over 0-1 losses corresponds to an agnostic weak online learner with advantage parameter $\gamma = \frac{2\eta}{1-e^{-\eta}}$ and regret $R_{\mathcal{W}}(T,k) = O(kd\log(\frac{1}{\delta}))$. Taking the discretization parameter $\delta = \frac{1}{T}$ ensures that the regret remains sublinear in $T$.

First, recall that for 0-1 losses $\tilde{L}_{y_t}(z_t) = \mathbb{1}\{z_t \neq y_t\}$, the EWA algorithm $\mathcal{W}$ using hypothesis class $\mathcal{H}$ and learning rate $\eta > 0$ guarantees the regret bound

$$\mathbb{E}\left[\sum_{t=1}^{T} \tilde{L}_{y_t}(\mathcal{W}(x_t))\right] \leq \frac{\eta}{1-e^{-\eta}} \mathbb{E}\left[\min_{h \in \mathcal{H}} \sum_{t=1}^{T} \tilde{L}_{y_t}(h(x_t))\right] + O(\log|\mathcal{H}|).$$

Next, for any values of $\ell_1, .., \ell_T$ where $\ell_t \neq y_t$, observe that

$$\mathbb{E}\left[\sum_{t=1}^{T} L_{y_t,l_t}(\mathcal{W}(x_t))\right] \leq \mathbb{E}\left[\sum_{t=1}^{T} \tilde{L}_{y_t}(\mathcal{W}(x_t))\right]$$

and

$$\frac{1}{2}\mathbb{E}\left[\min_{h \in \mathcal{H}} \sum_{t=1}^{T} \tilde{L}_{y_t}(h(x_t))\right] \leq \mathbb{E}\left[\min_{h \in \mathcal{H}} \sum_{t=1}^{T} L_{y_t,\ell_t}(h(x_t))\right].$$

Putting things together, we get

$$\mathbb{E}\left[\sum_{t=1}^{T} L_{y_t,l_t}(\mathcal{W}(x_t))\right] \leq \frac{2\eta}{1-e^{-\eta}} \mathbb{E}\left[\min_{h \in \mathcal{H}} \sum_{t=1}^{T} L_{y_t,\ell_t}(h(x_t))\right] + O(\log|\mathcal{H}|).$$

Finally, observing that $|\mathcal{H}| = \left(\frac{1}{\delta}\right)^{kd}$ completes the proof. As an example, if one takes $\eta = 1$, then the EWA algorithm corresponds to an agnostic weak online learner with advantage parameter $\gamma = \frac{2}{1-e^{-1}} \approx 3.16$ and regret $R_{\mathcal{W}}(T,k) = O(kd\log T)$.

A similar procedure can be performed when considering the same multiclass classification setup when the hypothesis class $\mathcal{H}$ is the set of depth 1 multiclass decision trees. If one further restricts to a $k$-wise split on a feature $j \in [d]$, then the class of depth 1 decision trees can be abstractly represented by the function:

$$f(x) = \begin{cases} y_1, & \text{if } -\infty < x_j \leq \tau_1 \\ y_2, & \text{if } \tau_1 < x_j \leq \tau_2 \\ ... \\ y_k, & \text{if } \tau_{k-1} < x_j < \infty \end{cases}$$

for input $x \in \mathbb{R}^d$, thresholds $\tau_1, ..., \tau_{k-1}$, and labels $y_1, ..., y_k$. If one discretizes the thresholds using the parameter $\delta \in (0,1)$, then $|\mathcal{H}| = O(d \cdot k^k \cdot (\frac{1}{\delta})^k)$. Therefore, EWA using this hypothesis class and 0-1 losses corresponds to an agnostic weak online learner with regret $R_{\mathcal{W}}(T,k) = O(k\log(kT) + \log d)$.

### D.2  Infinite Hypothesis Classes

Our construction of weak learners for the two learning settings above crucially relied on the fact that the hypothesis class was finite. Below, we discuss how to construct agnostic weak online learners for infinite hypothesis classes. The key insight from the constructions above is that any agnostic weak online learner with advantage $\gamma > 1$ for the standard 0-1 loss can be converted into weak learner that satisfies an equivalent version of Definition 1 with the non-negative loss functions $L_{y,l}(z)$ with twice the advantage, namely $2\gamma$. For finite hypothesis classes, we constructed weak learners for the 0-1 loss by running (randomized) EWA over $\mathcal{H}$ with a fixed learning rate $\eta$. Similarly, for infinite hypothesis classes, we can construct a weak learner for 0-1 losses by setting the learning rate $\eta$ to be a fixed

positive constant in the optimal multiclass agnostic online learner proposed in Daniely et al. [12]. Concretely, the multiclass agnostic online learner proposed in Section 5 of Daniely et al. [12] runs the (randomized) EWA over a carefully constructed finite set of experts of size at most $(Tk)^{LDim(\mathcal{H})}$, where $LDim(\mathcal{H})$ is the Multiclass Littlestone Dimension of $\mathcal{H}$. Following an identical analysis as in Daniely et al. [12], one can now show that if $\mathcal{W}$ is a EWA algorithm, then for 0-1 losses over this carefully constructed set of experts, $\mathcal{W}$ achieves regret

$$\mathbb{E}\left[\sum_{t=1}^{T} \tilde{L}_{y_t}(\mathcal{W}(x_t))\right] \leq \frac{\eta}{1 - e^{-\eta}} \mathbb{E}\left[\min_{h \in \mathcal{H}} \sum_{t=1}^{T} \tilde{L}_{y_t}(h(x_t))\right] + O(LDim(\mathcal{H})\log(Tk)),$$

where $\mathcal{H}$ is the hypothesis class of interest and $\eta > 0$. Picking $\eta = 1$, we get that the EWA $\mathcal{W}$ is an agnostic online weak learner for 0-1 losses with advantage parameter $\approx 1.58$. Thus, using the key insight mentioned above, $\mathcal{W}$ is also a agnostic online weak learner for the non-negative loss functions $L_{y,l}(z)$ with advantage parameter $\approx 3.16$. This suffices to show that $\mathcal{W}$ is also an agnostic online weak learner with respect to Definition 1.

# E   Important Lemmas

We give the important lemmas (and their proofs) used in the main text and Appendix A and B.

**Lemma 15.** *For any $t$ and $i$, $\mathbb{E}\left[y_t^i \cdot \mathcal{W}_i(x_t)\right] = \mathbb{E}\left[p_t^i \cdot \mathcal{W}_i(x_t)\right]$.*

*Proof.* First note that $\mathbb{E}\left[y_t^i | p_t^i, y_t\right] = p_t^i$. Next note that $\mathcal{W}_i(x_t)$ and $y_t^i$ are conditionally independent given $p_t^i$ and $y_t$. Then,

$$\begin{aligned}
\mathbb{E}\left[y_t^i \cdot \mathcal{W}_i(x_t)\right] &= \mathbb{E}_{p_t^i, y_t}\left[\mathbb{E}\left[y_i^t \cdot \mathcal{W}_i(x_t) | p_t^i, y_t\right]\right] && \text{(law of total expectation)} \\
&= \mathbb{E}_{p_t^i, y_t}\left[\mathbb{E}\left[y_t^i | p_t^i, y_t\right] \cdot \mathbb{E}\left[\mathcal{W}_i(x_t) | p_t^i, y_t\right]\right] \\
&= \mathbb{E}_{p_t^i, y_t}\left[p_t^i \cdot \mathbb{E}\left[\mathcal{W}_i(x_t) | p_t^i, y_t\right]\right] \\
&= \mathbb{E}_{p_t^i, y_t}\left[\mathbb{E}\left[p_t^i \cdot \mathcal{W}_i(x_t) | p_t^i, y_t\right]\right] \\
&= \mathbb{E}\left[p_t^i \cdot \mathcal{W}_i(x_t)\right].
\end{aligned}$$

$\square$

**Lemma 16.** *For any pair $h, y \in \mathcal{B}_k$ and $p \in \Delta_k$,*

$$p \cdot (2h - 2y) \geq 2(h \cdot y - 1)$$

*Proof.* Consider the case when $h = y$. Then the left and right side of the above inequality hold with equality at the value of 0 for any vector $p$ in the simplex. Now, consider the case when $h \neq y$. Then, observe that setting $p = y$ achieves the minimum of $p \cdot (2h - 2y)$ at $-2$ which matches the right-hand side. $\square$

**Lemma 17.** *For every $0 < \gamma \leq 1$, $h \in \Delta_k$, and $y \in \mathcal{B}_k$, there exists $p \in \Delta_k$ such that:*

$$p \cdot \left(\frac{2h - \mathbb{1}_k}{\gamma} - (2y - \mathbb{1}_k)\right) \leq 2\left(\prod\left(\frac{h}{\gamma}\right) \cdot y - 1\right),$$

*where $\prod : \mathbb{R}^k \to \Delta_k$ is the $L_2$ projection operator onto the simplex.*

*Proof.* We first review an important property of the $L_2$ projection onto the simplex. It is known that the solution to the $L_2$ projection $\prod(\cdot)$ onto the simplex is just a threshold operator (see Theorem 2.2 in [11]): for any input vector $g$, subtract a constant $\mu$ from each element. If the result is negative, replace it by zero. More formally, $\prod(g) = [g - \mu \mathbb{1}_k]_+$. Note that $\mu$ must satisfy the piecewise linear equation $\sum_{i=1}^k \max(0, g_i - \mu) = 1$. Let $g'$ denote the sorted version of $g$ in decreasing order. Then, define $K$ as the largest integer within $\{1, 2, ..., k\}$ such that $g'_K - \frac{\sum_{j=1}^K g'_j - 1}{K} > 0$. One can verify that $\mu = \frac{\sum_{j=1}^K g'_j - 1}{K}$ is the unique solution to the piece-wise linear equation above. We will use this property of $L_2$ projection extensively in our proof, which we formally begin below.

Define $\tilde{h} = \prod\left(\frac{h}{\gamma}\right)$. Let $t$ be the number of zero entries in the projection $\tilde{h}$. Furthermore, define $h_y = h \cdot y$ and $\tilde{h}_y = \tilde{h} \cdot y$ as the $y$'th index of $h$ and $\prod(\frac{h}{\gamma})$ respectively. We will show that when $t \le k - 2$, setting $p = y$ achieves the desired bound and when $t = k - 1$, there exists some $p \in \Delta_k$ that achieves the desired bound.

We now begin with the case where $t \le k - 2$. Our goal will be to first show the following lower bound on $\tilde{h}_y$,

$$\tilde{h}_y \ge \frac{(k-t)h_y - 1}{\gamma(k-t)} + \frac{1}{k-t}.$$

Consider the subcase where $\tilde{h}_y = 0$. Then, by the properties of the projection operator above it must have been the case that $\frac{h_y}{\gamma} \le \mu$, where using the definition above $\mu = \frac{\frac{1}{\gamma}\sum_{j=1}^{k-t} h'_j - 1}{k-t}$. Therefore, $h_y \le \frac{\sum_{j=1}^{k-t} h'_j - \gamma}{k-t} \le \frac{1-\gamma}{k-t}$. Substituting in, we find

$$\frac{(k-t)h_y - 1}{\gamma(k-t)} + \frac{1}{k-t} \le 0 = \tilde{h}_y.$$

Now, consider the subcase where $\tilde{h}_y > 0$. Then, $\tilde{h}_y = \frac{h_y}{\gamma} - \mu$, where again

$$\mu = \frac{\frac{1}{\gamma}\sum_{j=1}^{k-t} h'_j - 1}{k-t} \le \frac{1-\gamma}{\gamma(k-t)}.$$

Substituting in completes proving the lowerbound

$$\tilde{h}_y = \frac{h_y}{\gamma} - \mu \ge \frac{h_y}{\gamma} - \frac{1-\gamma}{\gamma(k-t)} = \frac{(k-t)h_y - 1}{\gamma(k-t)} + \frac{1}{k-t}.$$

Now that we have shown,

$$\tilde{h}_y = \frac{h_y}{\gamma} - \mu \ge \frac{(k-t)h_y - 1}{\gamma(k-t)} + \frac{1}{k-t},$$

we finally show this implies the inequality

$$\left(\frac{2h_y - 1}{\gamma} - 1\right) \le 2\tilde{h}_y - 2,$$

which follows from plugging $p = y$ into the lemma. We start with

$$2\tilde{h}_y - \frac{2h_y - 1}{\gamma} - 1 \geq \frac{2(k-t)h_y - 2}{\gamma(k-t)} + \frac{2}{k-t} - \frac{2h_y - 1}{\gamma} - 1$$

$$= -\frac{2}{\gamma(k-t)} + \frac{1}{\gamma} + \frac{2}{k-t} - 1$$

$$= (\frac{1}{\gamma} - 1) - \frac{2}{k-t}(\frac{1}{\gamma} - 1)$$

$$= (\frac{1}{\gamma} - 1)(1 - \frac{2}{k-t}) \geq 0,$$

where the last inequality follows from the assumption that $t \leq k - 2$. Thus, for the case where $t \leq k - 2$, we have shown the lemma holds by setting $p = y$.

Now we will consider the case where $t = k - 1$. Again, we will also consider the subcases where $\tilde{h}_y = 0$ and $\tilde{h}_y = 1$. We start with the subcase where $\tilde{h}_y = 0$. Here, we will show that picking $p = y$ is the right choice. Namely, we will show that the following inequality holds

$$\left( \frac{2h_y - 1}{\gamma} - 1 \right) \leq 2\tilde{h}_y - 2.$$

Under the assumption that $\tilde{h}_y = 0$, the right hand side collapses to $-2$. Thus, we need to show that $\frac{2h_y - 1}{\gamma} \leq -1$. Recall that if $\tilde{h}_y = 0$, then by projection properties. $\frac{h_y}{\gamma} \leq \mu$. Again, by definition,

$$\mu = \frac{\frac{1}{\gamma}\sum_{j=1}^{k-t} h'_j - 1}{k - t} = \max_j \frac{h_j}{\gamma} - 1 \leq \frac{1 - h_y}{\gamma} - 1.$$

Therefore we find that, $\frac{h_y}{\gamma} \leq \frac{1-h_y}{\gamma} - 1$, from which it is easy to see that $\frac{2h_y - 1}{\gamma} \leq -1$, which completes this subcase.

We now move to the subcase where $\tilde{h}_y = 1$. Here, we will show that there exists a $p \in \mathcal{B}_k \setminus \{y\}$ that satisfies the required bound in the lemma. When $\tilde{h}_y = 1$, the right hand side collapses to $0$. Thus, we need to show the existence of $p \in \mathcal{B}_k \setminus \{y\}$ s.t.

$$\frac{2h_p - 1}{\gamma} + 1 \leq 0,$$

where $h_p$ denotes $h \cdot p$, the value of $h$ at the $p$'th index. We shall prove the bound above via contradiction. Assume that indeed for all $p \in \mathcal{B}_k \setminus \{y\}$,

$$\frac{2h_p - 1}{\gamma} + 1 > 0.$$

This implies that $h_p > \frac{1-\gamma}{2}$ for all $p \in \mathcal{B}_k \setminus \{y\}$. Observe that when $k \geq 3$, the probability mass over all labels other than $y$ is *strictly* bounded below by $1 - \gamma$ and so it must be that $h_y < \gamma$. Recall that if $\tilde{h}_y = 1$, then by projection properties,

$$\mu = \frac{\frac{1}{\gamma}\sum_{j=1}^{k-t} h'_j - 1}{k - t} = \frac{h_y}{\gamma} - 1 < 0,$$

where the last inequality follows from the fact that $h_y$ must have mass strictly less than $\gamma$. However, if $\mu < 0$, then since the solution to the projection operation is $\left[ \frac{h}{\gamma} - \mu \mathbb{1}_k \right]_+$ and the entries of $h$ are non-negative, all entries of $\frac{h}{\gamma} - \mu \mathbb{1}_k$ are strictly positive. This contradicts our original assumption that $t = k - 1$ (implying that all but one index are negative) which completes the proof for $k \geq 3$.

Now, when $k = 2$, $\mu = \frac{h_y}{\gamma} - 1$, and by projection properties,

$$\frac{h_p}{\gamma} - \mu = \frac{h_p - h_y}{\gamma} + 1 \leq 0.$$

Noting that $h_y = 1 - h_p$, completes the proof since it implies that for $p \neq y$, $\frac{2h_p - 1}{\gamma} + 1 \leq 0$. $\quad\square$

**Lemma 18.** *For every $0 < \gamma \leq 1$, $h \in \Delta_k$, and $y \in \mathcal{B}_k$, there exists $p \in [0, 1]$ such that:*

$$p\left(\frac{2h \cdot y - 1}{\gamma} - 1\right) \leq 2\left(\prod\left(\frac{h}{\gamma}\right) \cdot y - 1\right),$$

*where $\prod : \mathbb{R}^k \to \Delta_k$ is the $L_2$ projection operator onto the simplex.*

*Proof.* Observe that we can consider the same four cases as in the proof for Lemma 17. Indeed, for three out of the four cases, the optimal choice for Lemma 17 was selecting $p = y$. Under these three cases, we know that by picking $p = y$ the following inequality holds by substituting $p = y$ into Lemma 17:

$$\frac{2h \cdot y - 1}{\gamma} - 1 \leq 2\left(\prod\left(\frac{h}{\gamma}\right) \cdot y - 1\right).$$

Thus, for this lemma, for the same three cases of Lemma 17, it suffices to pick $p = 1$. In the last case of Lemma 17, $\tilde{h}_y = 1$, that is, all the mass after the projection falls on the label $y$. In this case, the right hand side of our inequality collapses to 0. Thus it suffices to pick $p = 0$ for this lemma to hold. $\quad\square$

**Lemma 19.** *For $0 < \gamma \leq 1$, $h \in \Delta_k$, $y \in \mathcal{B}_k$, and all $p \in \Delta_k$:*

$$p \cdot \left(\frac{2h - \mathbb{1}_k}{\gamma} - (2y - \mathbb{1}_k)\right) = (2p - \mathbb{1}_k) \cdot \left(\frac{h}{\gamma} - y\right)$$

*Proof.*

$$\begin{aligned}
p \cdot \left(\frac{2h - \mathbb{1}_k}{\gamma} - (2y - \mathbb{1}_k)\right) &= p \cdot \left(\frac{2h - \mathbb{1}_k}{\gamma}\right) - p \cdot (2y - \mathbb{1}_k) \\
&= 2p \cdot \frac{h}{\gamma} - \frac{1}{\gamma} - 2p \cdot y + 1 \\
&= 2p \cdot \left(\frac{h}{\gamma} - y\right) - \frac{1}{\gamma} + 1 \\
&= 2p \cdot \left(\frac{h}{\gamma} - y\right) - \mathbb{1}_k \cdot \left(\frac{h}{\gamma} - y\right) \\
&= (2p - \mathbb{1}_k) \cdot \left(\frac{h}{\gamma} - y\right).
\end{aligned}$$

$\quad\square$

## F   Experimental Details

Each cell in Table 1 is the average over the "best accuracies" of five independent shuffles of the dataset. For each shuffle, the "best accuracy" was computed as the maximum accuracy over five candidate $\gamma$ values. For both OCO-based boosting algorithms, $\gamma$ was tuned across $[0.10, 0.30, 0.50, 0.70, 1]$, and for OnlineMBBM, $\gamma$ was tuned across $[0.001, 0.01, 0.05, 0.1, 0.3]$. The range of $\gamma$ values used to tune OnlineMBBM is consistent with those used by Jung et al. [24]. In addition, in our own experiments, we found that for $\gamma$ values larger than 0.30, OnlineMBBM performed significantly worse. Note that

Table 2: Dataset summaries.

| Dataset | #datapoints | #covariates | #classes |
|---|---|---|---|
| Balance | 625 | 4 | 3 |
| Cars | 1728 | 6 | 4 |
| LandSat | 6435 | 36 | 6 |
| Segmentation | 2310 | 19 | 7 |
| Mice | 1080 | 82 | 8 |
| Yeast | 1483 | 8 | 10 |
| Abalone | 4177 | 8 | 28 |

Table 3: Standard errors of accuracies.

| Dataset | Standard Errors of Accuracies | | | |
|---|---|---|---|---|
| | Agn | Opt | Ada | OCOR |
| Balance | 0.68 | 1.06 | 1.27 | 0.84 |
| Cars | 0.46 | 0.02 | 0.55 | 0.32 |
| LandSat | 0.89 | 0.36 | 1.23 | 0.21 |
| Segmentation | 0.70 | 0.48 | 1.56 | 0.54 |
| Mice | 0.58 | 0.60 | 2.53 | 0.26 |
| Yeast | 0.43 | 1.03 | 1.09 | 1.53 |
| Abalone | 0.53 | 0.42 | 0.28 | 0.31 |

AdaBoost.OLM does have not have a tuning parameter $\gamma$, which is one of its advantages despite its subpar performance.

For both OCO-based boosting algorithms, we used Online Gradient Descent with learning rate $\eta = \frac{\gamma}{\sqrt{N}}$ as the OCO algorithm. This learning rate is optimal up to constant factors [35].All computations were carried out on a Nehalem architecture 10-core 2.27 GHz Intel Xeon E7-4860 processors with 25 GB RAM per core. The total amount of computing time was around 500 hours. The River package [29], used to implement the VeryFastDecisionTree, is licensed under BSD 3-Clause.

Table 2 provides more information about each of the datasets used. For the Mice dataset, entries with missing data were replaced with the average value of their respective column.

Table 3 provides the standard error for each accuracy in Table 1.