# OpenReview forum: "Online Agnostic Multiclass Boosting"
_NeurIPS.cc/2022/Conference — NeurIPS 2022 Accept_

### Official Review · Reviewer_Xr8z · 2022-07-10

**Rating:** 7
**Confidence:** 3
**Soundness:** 3 good
**Presentation:** 3 good
**Contribution:** 3 good

**Summary:**

The main contribution of the paper is to introduce the first algorithm
for online agnostic multiclass boosting. In addition, it also presents
algorithms for multiclass boosting in the offline setting and the online
realizable setting, for which algorithms were already available.


**Questions:**

* Could you give some intuition why the definition of ell_t^i can have an
  arbitrary part? Isn't there some choice that is somehow best?
* The definition of weak learners depends on ell_t^i, but how is it
  chosen? Shouldn't it be passed from Algorithm 1?


**Limitations:**

* Although the theory is solid, and sufficient to support the paper, the
  empirical performance of the new algorithms in the experiments is not
  entirely convincing. (I do not consider this a mayor issue, but it
  suggests there may be room for fine-tuning or further investigation in
  future work.)


**Strengths And Weaknesses:**

This is a well written paper, which is the first to address the important issue of online agnostic multiclass boosting.

Proofs are clean and elegant.


Remarks:
* A strange aspect of the proof of Theorem 1 is that the definition of
  ell_t^i in line 185 is arbitrary in the second case. I have carefully
  checked the proof, though, and it seems fully correct.
* In line 172, the paper suggests to use online gradient descent as a
  learning algorithm over the probability simplex, but using
  Hedge would improve the dependence on k from O(sqrt{k}) to
  O(sqrt{ln(k)}).
* Experiments: it seems difficult to draw any conclusions from the
  reported accuracies in Table 1, which seem to vary greatly between
  data sets. In particular, the claim that "Compared to OnlineMBBM, our
  OCO-based boosting algorithms achieve comparable if not better
  performance" does not seem sufficiently supported, because it is only
  true if we compare the best of the two new algorithms to OnlineMBBM.

Minor comments:

* I found the choice of identifying integers ell in [k] with basis
  vectors e_ell, with the meaning depending on context, to be quite
  inconvenient. It works if the reader only wants to scan for global
  meaning, but for checking proofs it is an extra hurdle.
* I would suggest to change the notation for the predictions of the weak
  learners from cal{W}_i to cal{W}_t^i, for consistency with e.g. p_t^i.
* The most difficult parameter of Algorithm 1 is gamma, so it should be
  listed as an input.
* Line 176: O(k sqrt{T}) should be O(k sqrt{T}/gamma)
* Definition 2: Define cal{W}_{S'}, and "independent sample OF size m_0"

---

> ### Author Response · Authors · 2022-07-31
> **Response to Reviewer Xr8z  Part 1**
>
> **Q1**: *“In line 172, the paper suggests to use online gradient descent as a learning algorithm over the probability simplex, but using Hedge would improve the dependence on k from O(sqrt{k}) to O(sqrt{ln(k)})”*
>
> **A1**: We thank the reviewer for this comment. By Lemma 17 in  Appendix E, $l_t^i(p) = p \cdot ((2W-1)/\gamma - (2y-1)) = (2p-1) \cdot (W/\gamma - y)$. Thus, we can replace the loss function on line 9 of Algorithm 1 with $l_t^i(p) = (2p-1) \cdot (W/\gamma - y)$, without changing the outputs of the OCO algorithm. Indeed, one can verify that the iterates of the OCO algorithm after projection remain the same regardless of whether one uses $l_t^i(p) = p \cdot ((2W-1)/\gamma - (2y-1))$ or $l_t^i(p) = (2p-1) \cdot (W/\gamma - y)$. Now, note that the upper bound on the gradient of $(2p-1) \cdot (W/\gamma - y)$ actually does not depend on $k$ at all. Thus, the only dependence on $k$ is with respect to the regret of the weak learner.  We will make this more explicit in the final version of the paper.
>
> **Q2**: *“In particular, the claim that "Compared to OnlineMBBM, our OCO-based boosting algorithms achieve comparable if not better performance" does not seem sufficiently supported, because it is only true if we compare the best of the two new algorithms to OnlineMBBM.”*
>
> **A2**: We thank the reviewer for pointing this out. However, there is only one dataset (Landsat) where the minimum of the two OCO-based boosting algorithms is significantly underperforming OnlineMBBM. In addition, we agree with the reviewer that based on our experimental results, there is no clear winner. However, we do find that our OCO-based boosting algorithms are comparable to existing boosting algorithm with the edge of also being faster.
>
> **Q3**: *“I found the choice of identifying integers $\ell in [k]$ with basis vectors $e_\ell$, with the meaning depending on context, to be quite inconvenient. It works if the reader only wants to scan for global meaning, but for checking proofs it is an extra hurdle.”*
>
> **A3**: We thank the reviewer for pointing out this concern. In the final version, we will make the notation more consistent by just using $e_i$ to reference the $i$th basis vector instead of abusing notation.
>
> **Q4**: *“I would suggest to change the notation for the predictions of the weak learners from $W_i$ to $W_t^i$, for consistency with e.g. $p_t^i$”*
>
> **A4**: We thank the reviewer for this comment. Unfortunately, this would imply that there are $NT$ weak learners, when in fact there are only $N$ weak learners, each of which make predictions and update their internal state $T$ times. In contrast, there are actually $NT$ values for $p_t^i$. The weak learner’s prediction at time $t$ can be inferred from the fact that it is receiving as input example $x_t$ (example is indexed by $t$). This notation was also used by Brukhim et al. in the binary case.
>
> **Q5**: *“The most difficult parameter of Algorithm 1 is gamma, so it should be listed as an input”*
>
> **A5**: We thank the reviewer for pointing this out and will include gamma in the final version.
>
> **Q6**: *“Line 176: O(k sqrt{T}) should be O(k sqrt{T}/gamma)”*
>
> **A6**: We thank the reviewer for pointing this out and will make the suggested change in the final version.
>
> **Q7**: *“Definition 2: Define $W_{S'}$, and "independent sample OF size $m_0$"*
>
> **A7**: We thank the reviewer for pointing these out. We will make these changes in the final version.
>
> **Q8**: *“Could you give some intuition why the definition of $\ell_t^i$ can have an arbitrary part? Isn't there some choice that is somehow best?”*
>
> **A8**: We thank the reviewer for this question. When the weak learner is correct, regardless of what $\ell_t^i$ is, the gain of the weak learner is 1. So from the Booster’s perspective, the value of $\ell_t^i$ does not matter when the weak learner is correct.  Thus, in this case the Booster can select this value arbitrarily.

---

> ### Author Response · Authors · 2022-07-31
> **Response to Reviewer Xr8z Part 2**
>
> **Q9**: *“The definition of weak learners depends on $\ell_t^i$, but how is it chosen? Shouldn't it be passed from Algorithm  1?”*
>
> **A9**: We thank the reviewer for pointing out this subtle detail. The reference label is needed for the *analysis* but is not needed in the *algorithm* since it is not actually set by the Booster or observed by the weak learner. However, one could modify Algorithm 1 to make this more explicit by having the Booster construct $\ell_t^i$ as defined in line 185, and pass $(x_t, y_t^i, l_t^i)$ to weak learner $i$ in round $t$. However, note that a weak learner needs only the example and its label to update its internal state, the reference label only controls how much loss the weak learner suffers when its wrong - it doesn’t actually help it learn the optimal hypothesis. Thus, we make a distinction between the gain that the weak learner observes internally, and the gain that the Booster assigns the weak learner. While the weak learner gains $[-1, 1]$  reward internally, externally and oblivious to its own self, the Booster sets the weak learners gain to be $[-1, 0, 1]$ based on a chosen reference label. This is exactly why the weak learner does not need to observe the reference label $\ell_t^I$.
>
> **Q10**: *"Although the theory is solid, and sufficient to support the paper, the empirical performance of the new algorithms in the experiments is not entirely convincing. (I do not consider this a mayor issue, but it suggests there may be room for fine-tuning or further investigation in future work.")*
>
> **A10**: We thank the reviewer for this comment. We would like to remark that the main focus of our paper was on the theoretical side. We do agree that our experimental results don't point to a clear winner amongst the Boosting algorithms implemented. That said, we do find that our OCO-based boosting algorithms are comparable to existing algorithms with the advantage of also being fast and easy to implement. We lastly agree that a thorough investigation into empirical performance of OCO-based boosting algorithms will be an interesting future direction.

---

> > ### Comment · Reviewer_Xr8z · 2022-08-08
> > **About the definition of a weak learner**
> >
> > Dear authors,
> >
> > Thanks for your response.
> >
> > I do not fully understand your answer that the weak learner does not observe the reference labels $\ell_t^i$, because they seem to be a crucial part of measuring its performance in Definition 1, and they are also used in the example constructions of weak learners in Appendix D.
> >
> > However, since there is no issue of correctness, I will not ask for further clarification, which will not change my score anyway.

---

### Official Review · Reviewer_AAXr · 2022-07-11

**Rating:** 5
**Confidence:** 5
**Soundness:** 3 good
**Presentation:** 3 good
**Contribution:** 2 fair

**Summary:**

This work proposes an extension of online agnostic boosting techniques to the multiclass setting, via a reduction to OCO. They further generalize the approach to statistical agnostic, online realizable, and statistical realizable multiclass boosting. They give several WL conditions, and provide corresponding bounds for each condition. Lastly, they give an empirical evaluation of their approach.


**Questions:**


1. Figure 1 - is not very clear. Consider adding description of the axis, and further explain how to interpert the plot. Maybe explicitly say that the black,blue,red lines correspond to Delta_k/gamma for the different gammas?

2. Why is “L2 projection is not a natural projection operator for Delta_k”? not clear why it is needed to first project to the line that contains Delta_k, and only then to Delta_k - why not directly project to Delta_k?

3. Weak learning assumption : regarding Definition 1, the assumption that W performs well w.r.t this sequence of ell_t without ever seeing it seems quite strong. In addpendix D the example given which satisifies this requirement seems to also be irelevant for boosting, i.e., for the setting specified in appendix D applying the boosting procedure only worsen the regret bound, compared to simply running randomized weighted majority ?

4. Missing citations: the paper does not discuss their approach in comparison to other online boosting framworks, apart from Brukhim et al. 2020. It would be helpful to compare your result to e.g., “Online Gradient Boosting” by Beygelzimer, Hazan, Kale, Luo, 2015, and “Online Boosting with Bandit Feedback” by Brukhim, Hazan, 2021.  Both of these works are not restricted to binary labels or realizability assumptions. Also consider discussing the works “Optimal and adaptive algorithms for online boosting” Beygelzimer, Kale, Luo 2015, and and “Online Multiclass Boosting” by Jung, Goetz, Tewari, 2017. - and why they do not easily extend to the multiclass agnostic setting?

5. Regarding the alternative definitions in appendix C: Def. 5 is too strong as well, as th authors pointed out. Def. 6 is unclear to me - does the learner W get to observe the label \ell ? it seems like it doesn’t, and that also seems to be a strong assumpution. If we consider extending a learnable class H to also contain all k fixed-label functions (this does not change the complexity of H), then the average correlation of the learner per label is at least gamma, for all labels simoultanously.


6. Regarding appendix C in general, no one of the results mentioned there are proved. I understand this is derived similarly to the previous results, consider adding a formal proof for clarity as it is not entirely trivial.


7. Mukherjee an Schapire 2013 discuss various WL conditions for multiclass boosting in the batch setting. It would be beneficial to at least include some discussion on the various conditions given in the paper and their relative strengths and weaknesses, as this seems to be a main contribution of this work.


8. Regarding appendix D,  what is the expectation taking with respect to? in the statemnt of Def.1 you say that \ell_t may depend on W(x_t), so it is randomized and cannot be fixed beforehand (though the inequalities still holds regardless).
    - (also a minor comment - please refer to a specific source for the REWA algorithm and a regret bound guarantee or include it in the writeup, since there are many variations and naming conventions so it is not clear to less familiar readers to which one you are refrring to).

 9. Lemma 14 seems to be a critical component of the overall proof, which is given via a case-by-case analysis. It would be interesting to also have a high-level intuitive explanation for why this works?


**Limitations:**

There is no potential negative societal impact of the work.

**Strengths And Weaknesses:**

**Strengths**

1. This approach seems to be a natural extension of previous boosting methods to the online multiclass setting. The analysis also seems to be rigorous.
2. The various weak learning conditions given in this work and their corresponding bounds under the same boosting framework is an interesting contribution.

**Weaknesses**

1. The novelty looks limited as the techniques are very similar to Brukhim et al. 2020. Moreover, this paper does not discuss the major difficulty of extending that framework to the multiclass setting.
2. The WL assumptions seems to be quite strong (see details below). It would greatly improve the work to either relax these conditions or give a lower bound showing this cannot be done.

---

> ### Author Response · Authors · 2022-07-31
> **Regarding the WLC**
>
> We agree with the Reviewer that the weak learning condition in Definition 1 is a bit strong, as the adversary can pick $\ell_t$ even after observing the prediction of the weak learner. That said, we would also like to point out that we are the first to consider this difficult boosting setting. This comment serves to provide more detail/discussion into Definition 1 and what feasible online weak learning conditions should look like.
>
> We first identify two important requirements for any reasonable online weak learning condition in the agnostic multiclass setting: (1) the weak learning condition should reduce to the same condition posed by Brukhim et al. for binary classification and (2) the cumulative expected gain of a randomly guessing competitor should be non-negative. The second requirement is necessary in order for $\gamma \in (0, 1)$. Note that Definition 5 in Appendix C fails to satisfy condition (2). We have considered several possible weak learning conditions that satisfy both (1) and (2). However, after significant effort, we were ultimately only able to prove regret bounds for Definition 1, 6, and 7.
>
> While we agree Definition 1 is strong, we can show that it is not too hard to construct weak learners that satisfy it. Indeed, in Appendix D, we give examples of weak leaners that satisfy an equivalent version (affine transformation) of Definition 1 for finite hypothesis classes. From this construction, we make an important observation: any agnostic weak online learning with advantage $\gamma > 1$ for the standard 0-1 loss can be converted into weak learner that satisfies the equivalent version of Definition 1 with twice the advantage, namely $2\gamma$. Therefore, constructing a learner satisfying the weak learning condition for the equivalent version of Definition 1 is as hard as constructing a weak learning for the standard 0-1 loss. In Appendix D, we give examples of how to construct weak learners for the 0-1 loss for finite hypothesis. The key idea is to run the experts algorithm with a fixed learning rate. For any infinite hypothesis class, a similar weak learner for 0-1 losses can be constructed by setting $\eta$ to be a fixed constant (say $\eta$ = 2) in the “Learning with Expert Advice” algorithm in [1] (see page 21).
>
> 1. Amit Daniely, Sivan Sabato, Shai Ben-David, and Shai Shalev-Shwartz. Multiclass learnability and the erm principle. In COLT, pages 207–232, 2011.

---

> ### Author Response · Authors · 2022-07-31
> **Response to Reviewer  AAXr Part 1**
>
> **Q1**: *“The novelty looks limited as the techniques are very similar to Brukhim et al. 2020. Moreover, this paper does not discuss the major difficulty of extending that framework to the multiclass setting”*
>
> **A1**: We thank the reviewer for this comment. Although we do build on the work by Brukhim et al. 2020, there are several difficulties that arise when attempting to generalize to the multiclass setting. The first, and  most important, difficulty is identifying the correct weak learning condition for online muticlass agnostic boosting, which turned out to be far from trivial. The second difficulty is how to generalize the randomized-majority vote/projection to the multiclass setting. Indeed, it is not obvious that the L2 projection onto the simplex is the correct projection, and the proof of its correctness is non-trivial either. Third is the difficulty in identifying how exactly the Booster should update the weak learners in the multiclass setting. It is not obvious what convex space the OCO algorithm should operate over, what the appropriate loss function should be to enable the regret analysis, or how the booster should relabel examples using the output of the OCO. There are many different options for each of these, all of which can converge to the procedure used by Brukhim et al. in the binary setting. Lastly, we note that there is a delicate interplay between each of the aforementioned pieces. For example based on the proof of Theorem 1, it can be seen that both the weak learning condition and the randomized weighted majority prediction method directly impact what loss function we can/should set for the OCO. We do discuss some of these difficulties in the paper (see paragraph right below Alg 1), however, we will expand upon this in the final version.
>
> **Q2**: *“The WL assumptions seems to be quite strong (see details below). It would greatly improve the work to either relax these conditions or give a lower bound showing this cannot be done.”*
>
> **A2**: We thank the reviewer for pointing out that the WL assumptions are a bit strong. We agree with the reviewer and have created a separate comment ("Regarding the WLC") that discusses the strength of our weak learning condition.
>
> **Q3**: *“Figure 1 - is not very clear. Consider adding description of the axis, and further explain how to interpert the plot. Maybe explicitly say that the black,blue,red lines correspond to $\Delta_k/gamma$ for the different gammas?”*
>
> **A3**: We thank the reviewer for this suggestion and we will incorporate it in the final version, along with a description of the axis.
>
> **Q4**: *“Why is “L2 projection is not a natural projection operator for $\Delta_k$”? not clear why it is needed to first project to the line that contains $\Delta_k$, and only then to $\Delta_k$ - why not directly project to $\Delta_k$?”*
>
> **A4**: We thank the reviewer for this comment. For the probability simplex, we think the KL divergence is a more natural “metric” to measure distance between two points compared to the Euclidean distance metric. We note that the L2 projection onto the simplex is a direct projection onto the simplex. It takes a point in $R^d$ and maps it directly to a point in the simplex. However,  to better understand what this L2 projection is doing in our context of Boosting, and to provide some intuition on why it works, we found that it was useful to *view* it as the two step process described in Section 3.1 of the main text.
>
> **Q5**: *“Weak learning assumption : regarding Definition 1, the assumption that W performs well w.r.t this sequence of $\ell_t$ without ever seeing it seems quite strong. In addpendix D the example given which satisifies this requirement seems to also be irelevant for boosting, i.e., for the setting specified in appendix D applying the boosting procedure only worsen the regret bound, compared to simply running randomized weighted majority ?”*
>
> **A5**: We thank the reviewer for this comment. The point of the example in Appendix D is to (1) show it is not too hard to construct weak learners that satisfy Definition 1 and (2) show what the dependence of $R_W(T, k)$ on $k$ might look like.
>
> Even though for finite hypothesis classes, one can indeed  run randomized weighted majority to achieve sublinear regret, this requires $\eta$ to be tuned precisely to its optimal value. However, this ability to properly tune $\eta$ might not be always be feasible (e.g. for privacy reasons). In contrast, we can construct weak learners by setting $\eta$ apriori to any constant of our liking.

---

> ### Author Response · Authors · 2022-07-31
> **Response to Reviewer AAXr Part 2**
>
> **Q6**: *“Missing citations: the paper does not discuss their approach in comparison to other online boosting framworks, apart from Brukhim et al. 2020. It would be helpful to compare your result to e.g., “Online Gradient Boosting” by Beygelzimer, Hazan, Kale, Luo, 2015, and “Online Boosting with Bandit Feedback” by Brukhim, Hazan, 2021. Both of these works are not restricted to binary labels or realizability assumptions. Also consider discussing the works “Optimal and adaptive algorithms for online boosting” Beygelzimer, Kale, Luo 2015, and and “Online Multiclass Boosting” by Jung, Goetz, Tewari, 2017. - and why they do not easily extend to the multiclass agnostic setting?”*
>
> **A6**: We thank the reviewer for pointing out these missing citations and comparisons.  We will make sure to add a detailed discussion of these works and how they compare to our setting in the related works section of our final version.
>
> **Q7**: *“Regarding the alternative definitions in appendix C: Def. 5 is too strong as well, as th authors pointed out. Def. 6 is unclear to me - does the learner W get to observe the label $\ell$ ? it seems like it doesn’t, and that also seems to be a strong assumpution.”*
>
> **A7**: The weak learning condition in Definition 6 requires the regret bound to hold for all reference labels $\ell$ in hindsight. More precisely, Definition 6  requires that in hindsight, for every reference label $\ell$, the weak learner should achieve sublinear regret with respect to all examples labelled as $\ell$. We agree with the reviewer that this condition is a bit of a strong requirement. We ask the reviewer to see the separate comment we have made about the strength of our weak learning conditions.
>
> **Q8**: *“Regarding appendix C in general, no one of the results mentioned there are proved. I understand this is derived similarly to the previous results, consider adding a formal proof for clarity as it is not entirely trivial.”*
>
> **A8**: We thank the reviewer for this suggestion and will include the formal proofs in the final version. That said, the only difference in the proof is with respect to the lowerbound on the sum of the losses passed to the OCO algorithm, and specifically with respect to the equations on line 187 and 188.
>
> **Q9**: *“Mukherjee an Schapire 2013 discuss various WL conditions for multiclass boosting in the batch setting. It would be beneficial to at least include some discussion on the various conditions given in the paper and their relative strengths and weaknesses, as this seems to be a main contribution of this work.”*
>
> **A9**: We thank the reviewer for this suggestion and will do so accordingly in the final version. In Appendix C, we do have some preliminary comparisons of the weak learning conditions we propose, however, we will make sure to expand and add a more detailed discussion in the final version.
>
> **Q10**: *“Regarding appendix D, what is the expectation taking with respect to? in the statemnt of Def.1 you say that $\ell_t$ may depend on $W(x_t)$, so it is randomized and cannot be fixed beforehand (though the inequalities still holds regardless).”*
>
> **A10**: The expectation is taken with respect to only the weak learners predictions and the  adversaries selection of example-label pairs. That is, in Appendix D, we do not treat the reference labels as being random. However, note that the inequalities connecting the $[0, 1]$ loss with the $[0, 1/2, 1]$ loss holds pointwise for any selection of reference labels $l_1, …, l_t$ (whether they be random or not). Therefore, as you mentioned, their randomness as in the Definition 1, does not impact the claim.
>
> **Q11**: *“(also a minor comment - please refer to a specific source for the REWA algorithm and a regret bound guarantee or include it in the writeup, since there are many variations and naming conventions so it is not clear to less familiar readers to which one you are refrring to)”*
>
> **A11**: We thank the reviewer for this suggestion and we will make sure to do so in the final version. To clarify, by REWA, we are referring to the exponentially-weighted forecaster from Chapter 2 of [1], and its regret bound guarantee from Theorem 2.4 in Chapter 2 of [1].
>
> **Q12**: *“Lemma 14 seems to be a critical component of the overall proof, which is given via a case-by-case analysis. It would be interesting to also have a high-level intuitive explanation for why this works?”*
>
> **A12**: We agree that a more intuitive and geometric explanation for why the L2 projection operator works would be very interesting. We have attempted to provide some intuition in the main text (see Section 3.1 under **Randomized Prediction**) by showing that the L2 projection from a $\gamma$-scaled simplex to the simplex can be viewed as a two stage process, and then providing some intuition on what this means from the Booster's point of view.
>
> 1. N. Cesa-Bianchi and G. Lugosi. Prediction, Learning, and Games. Cambridge University Press, USA, 2006. ISBN 0521841089.

---

### Official Review · Reviewer_8xyp · 2022-07-12

**Rating:** 7
**Confidence:** 3
**Soundness:** 3 good
**Presentation:** 3 good
**Contribution:** 3 good

**Summary:**

The paper presents an extension of the results in Brukhim et al 2020 to multiclass problems, with similar extensions to online & batch settings for both adversarial and realizable problems.

**Questions:**

The agnostic setting allows the label generating function to be adversarial which is significantly more flexible, but there are several intermediate points between a stationary labelling function in the realizable case and an entirely adversarial labelling function in the agnostic case, which is a non-stationary but not adversarial labelling function (e.g. operating under concept drift). Are the bounds tighter in the non-stationary but not adversarial setting?

Does Algorithm 1 reduce to the algorithm of Brukhim et al 2020 in the binary case?

Why does the agnostic algorithm (agn) outperform the realizable version (ocor), when the datasets should be realizable?

**Limitations:**

The experimental study is small, and would have benefited from more complex datasets involving non-stationary or adversarial behaviour to make clear the distinction between the agnostic algorithm and the existing realizable ones.

**Strengths And Weaknesses:**

Overall the presentation of the material is good, and the topic is interesting.

This paper is rather full, with much of the supplementary material being necessary to understand Section 4, and the paper itself relying strongly on Brukhim et al 2020 for longer explanations of several of the concepts. It might be better as a longer journal paper with all the material in the appropriate order rather than forced into an overly terse 9 pages.

The modification from random relabelling to fractional relabelling is discussed very briefly in the experimental results, but this seems to be quite an important difference which should be covered in more detail. The fractional relabelling approach is approximating passing in the full distribution as a target, and if the weak learner was capable of fitting the full probability distribution (as for example a logistic regression trained under cross entropy would be) then passing in a single example with $y_t$ replaced by $p_t^i[l]$ seems like it would achieve the same goals more efficiently.

---

> ### Author Response · Authors · 2022-07-31
> **Response to Reviewer 8xyp**
>
> **Q1**: *“This paper is rather full, with much of the supplementary material being necessary to understand Section 4, and the paper itself relying strongly on Brukhim et al 2020 for longer explanations of several of the concepts. It might be better as a longer journal paper with all the material in the appropriate order rather than forced into an overly terse 9 pages.”*
>
> **A1**: We thank the reviewer for this comment. Our paper is mainly focused on just the online agnostic setting. With this in mind, most of the results (including proofs) for the online agnostic setting are included in the main paper, except for the additional lemmas in the Appendix.
>
> **Q2**: *“The modification from random relabelling to fractional relabelling is discussed very briefly in the experimental results, but this seems to be quite an important difference which should be covered in more detail."*
>
> **A2**: We thank the reviewer for this comment. This idea for fractional relabelling was borrowed from [1], which observed a similar phenomena empirically for binary classification.  We will clarify this further and add more details in the main text. A deeper understanding of this empirical phenomena is indeed an interesting future direction!
>
> **Q3**: *“Are the bounds tighter in the non-stationary but not adversarial setting?”*
>
> **A3**: We thank the reviewer for this question. We are unsure about the tightness in the non-stationary setting since in online learning it is standard to place no assumptions on the sequence of examples. The setting of online learning with non-stationary data is an interesting consideration for future work.
>
> **Q4**: *“Does Algorithm 1 reduce to the algorithm of Brukhim et al 2020 in the binary case?”*
>
> **A4**: We thank the reviewer for this question. Yes, Algorithm 1 does reduce to the algorithm of Brukhim et al. 2020 in the binary case. The weak learning condition in Definition 1 reduces directly. Furthermore, one can show that the exact same distribution $p_t^i$ over the labels is achieved by our algorithm and the one by Brukhim et al. in each round for each weak learner. We will further clarify this in the updated version.
>
> **Q5**: *“Why does the agnostic algorithm (agn) outperform the realizable version (ocor), when the datasets should be realizable?”*
>
> **A5**: We thank the reviewer for this comment. With our real-life datasets and choice of weak learners, it is actually not clear whether the realizability assumption is met for any of the datasets. Interestingly, we find that the performance of OnlineMBBM and OCOR often move together from one dataset to another, perhaps being an indicator of the realizability of a particular dataset.
>
> **Q6**: *“The experimental study is small, and would have benefited from more complex datasets involving non-stationary or adversarial behaviour to make clear the distinction between the agnostic algorithm and the existing realizable ones.”*
>
> **A6**: We thank the reviewer for this comment. We emphasize that the main focus of our work is theoretical. We agree that our experimental results do not indicate a clear winner and would benefit from more complicated datasets. However, even with the real-world datasets we used, it is unclear whether the realizability assumption holds for any of them, and thus they could potentially contain adversarial behavior. In addition, we do find that our OCO-based boosting algorithms are comparable to existing boosting algorithms with the added benefit of being faster and easier to implement.
>
> 1. V. Kanade and A. Kalai. Potential-based agnostic boosting. Advances in neural information processing systems, 22, 2009.

---

### Official Review · Reviewer_d9wh · 2022-07-21

**Rating:** 6
**Confidence:** 3
**Soundness:** 3 good
**Presentation:** 2 fair
**Contribution:** 2 fair

**Summary:**

In this paper, the authors introduce an agnostic online learning boosting algorithm for multiclass classification problems extending the prior work [5] for binary classification (N. Brukhim et al., NeurIPS-20). The algorithm is based on a generic reduction of boosting to online convex optimization (OCO) which was proposed in the aforementioned paper. Given access to agnostic weak online learners (AWOLs) satisfying some error bound, the algorithm constructs an online learner with a regret bound. The proposed procedure can be extended to other settings (statistical agnostic, online realizable, and statistical realizable) but does not lead to state-of-the-art regret bounds in these settings. Finally, the authors numerically compare the proposed agnostic and realizable online boosting algorithms with OnlineMBBM and Adaboost.OLM. The experimental results show that the proposed algorithms are competitive with existing multiclass boosting algorithms.

**Questions:**

The main setting considered in the paper is fully adversarial, not just agnostic: the environment adapts to the player. This entails a stronger condition on weak learners (see Definition 1) as compared to the same requirement in the non-adaptive case. At the same time, I believe, non-adaptive environment is more realistic for most practical cases. To what extent is the condition in Definition 1 more restrictive than that in the non-adaptive case? For example, are there generic examples for class of experts where the maximum of expected gain is at least positive? (as it required in ll. 518-520 of Supplementary?).

I would consider a much simpler and interpretable condition for AWOL in Definition 1 without auxiliary variables l_t. We can just use 0-1 loss as the gain function instead of strange \sigma_{y_t, l_t}. Such variant of Definition 1 has the following properties:
(1)	It is also a generalization of Definition 1 from [4]
(2)	It is strictly weaker than the current requirement to AWOL in Definition 1.
Does the regret bound work for this version of Definition 1?

“Importantly, we also allow the adversary to pick l_t even after it has observed W(x_t)”. Could you provide some intuition why do you want the environment to controls how much loss the weak learner suffers when it is incorrect?

**Limitations:**

I see not potential negative societal impact

**Strengths And Weaknesses:**

**Strengths**

The problem is well motivated and, in my opinion, is relevant to the NeurIPS community. The paper fills a gap in the literature by providing the multiclass boosting algorithm in the online agnostic setting.

Overall, the paper is well written. The structure of the paper is clear. All the stated claims are supported by proofs.

The generalization of online agnostic boosting from [4] (Brukhim et al., 2020) to multiclass problems looks nontrivial and requires careful analysis. Namely, new complications in the multiclass setting arise in construction of a loss function which is passed by the booster to an OCO algorithm. However, I expect some comment on how the algorithm correspond to that of  [4] in the particular case k=2. Given that it the algorithms have the same regret bounds, are they equivalent in some sense?

The paper provides more intuition behind the proposed algorithms than the original paper [Brukhim et al., 2020]. Additionally, the paper discusses how to construct AWOLs for two types of finite hypothesis classes: the set of discretized weight matrices and the set of discretized multiclass decision trees of depth 1. Although these examples are simple, they were not discussed in [Brukhim et al., 2020].

Finally, the authors conduct numerical experiments for the proposed methods.


**Weaknesses**

First of all, I have several questions about the problem setting and the significance of the obtained results (see below). My score depends on the answers.

Second, I would appreciate particular formulations of regret bounds for the adaptive environment as it announced in ll. 167-168. Currently, the main result assumes a strong requirement for weak learners (adaptive env.) and in a weak setting for boosting algorithm (non-adaptive env.). This makes the main result something inapplicable: it does not correspond to any of the two settings.

It is known that the reduction of boosting to OCO leads to suboptimal regret bounds when applied to realizable boosting and statistical agnostic boosting; see [Brukhim et al., 2020]. In Section 4, where other settings are discussed, the authors write that their purpose is “not to achieve state-of-the-art bounds for these settings”. I understand that the reduction to OCO may not lead to the optimal regret bounds but, however, it seems necessary to me to compare the obtained results with the existing ones. I would suggest, for instance, to move some of the proofs to the appendix and add this comparison.

Experiments. I understand that the main contribution of the paper is the theoretical results. However, in my opinion, experiment section can be improved.
(1) The number of weak learners is set to only 100, and simple decision trees of depth 1 are used as weak learners. In practice, real models are much more complicated. Is it true that the proposed algorithms are so complex that it is hard to apply them in more realistic settings. It would be useful to provide complexity analysis of the proposed algorithms.
(2) I am confused by the fact that the accuracy may differ by more than 10% within a single dataset. Is it a consequence of (1)?
(3) It is difficult to make any conclusions from experimental results, since more expensive algorithms generally achieve higher accuracy. A “fair” comparison requires to assign equal computational budget to all methods (all of them have hyperparameters, which allow to trade-off accuracy and training time).
(4) What parameters for Projected OGD are used? Why in OnlineMBBM the advantage parameter was tuned across [0.001, 0.01, 0.05, 0.1, 0.3]? Experimental part needs much more details.
(5) Only seven quite small datasets are used.
(6) It would be useful to have also some average accuracy over all datasets and an alternative measure (e.g., cross-entropy).

At last, formulations are not absolutely perfect:

- In equations for expected regret (l. 69-70) and AWOL condition (ll. 155-156), y_t / l_t should be different in the first and the second term of the differences, because y_t / l_t depend on the expert/algorithm. This is the consequence of the environment adaptivity, and the use of the same notations for different variables indicates, conversely, the non-adaptive case.

- In ll. 67-68, the dimension of the standard simplex is k-1, not k, see https://en.wikipedia.org/wiki/Simplex#The_standard_simplex
Please, respect general mathematics.

Minor comments:
(1) A small overview of OCO algorithms would be helpful for the reader, as OCO is a central part of the proposed algorithms. At the same time, the detailed proof of the main theorem can be moved to Suppl.
(2) Caption for Figure 1: the dot after p_2^* appears at the beginning of the second line.
(3) L187, second line, first term: factor 2 should be inside the bracket

---

> ### Author Response · Authors · 2022-07-31
> **Response to Reviewer d9wh Part 1**
>
> **Q1**: *“Second, I would appreciate particular formulations of regret bounds for the adaptive environment as it announced in ll. 167-168. Currently, the main result assumes a strong requirement for weak learners (adaptive env.) and in a weak setting for boosting algorithm (non-adaptive env.)”*
>
> **A1**:We thank the reviewer for pointing this out. As you mentioned, the weak learning assumption assumes an adaptive adversary, as is standard in agnostic online learning. Note even though we proved a regret bound for the Booster in the oblivious setting, we need a weak learning condition assuming an adaptive adversary because the weak learners observe randomly relabelled data. As you pointed out, the regret bound we prove for the Booster is assuming the oblivious setting. However, this same regret bound can be generalized *exactly* to the adaptive setting via standard arguments presented on pg. 69 in [1], specifically Lemma 4.1, as well as Exercise 4.1 on pg. 95. A key requirement allowing an oblivious regret bound to generalize to an adaptive regret bound is that the learner’s predictions on round $t$ should not depend on any of its past predictions from previous rounds. This is indeed true for our Booster. We will expand upon this further in the final version of the paper.
>
> **Q2**: *“I understand that the reduction to OCO may not lead to the optimal regret bounds but, however, it seems necessary to me to compare the obtained results with the existing ones.”*
>
> **A2**:We thank the reviewer for pointing this out. We will provide a comparison of our results with existing realizable and agnostic boosting algorithms in the final version, but perhaps in the Appendix. Note that, to our best knowledge, there are no comparable *agnostic* mutliclass boosting algorithms for both the online and batch setting. In the batch setting, Brukhim et al. in [2] do study agnostic multiclass boosting, however, as mentioned in the related works, their model is a bit different than ours. We will comment on this further in the final version of the paper.
>
> **Q3**: *“The number of weak learners is set to only 100, and simple decision trees of depth 1 are used as weak learners. In practice, real models are much more complicated. Is it true that the proposed algorithms are so complex that it is hard to apply them in more realistic settings. It would be useful to provide complexity analysis of the proposed algorithms”*
>
> **A3**: We thank the reviewer for this comment.  All our boosting algorithms can be used with any number and any kind of “weak” learner (even potentially complicated strong models). We chose $N = 100$ based on the number of weak learners selected in previous works (see [3]) and used depth-1 decision trees due to their simplicity and popularity. Again, we remark that any online algorithm can be used as a weak learner. However, with boosting, we typically lean towards simpler weak learners. With respect to complexity, all our boosting algorithms are efficient assuming access to an efficient weak learner. More specifically, for each round $t$, if the running time of the WL is $Q$, then the running time of Alg1 is $O(NQ + Nklogk)$. Note that the OnlineMBBM algorithm proposed in [3] by Jung et al. is not efficient even assuming access to an efficient weak learner. We will expand further about the running time of our OCO-based boosting algorithms in the final version of paper.
>
> **Q4**: *“I am confused by the fact that the accuracy may differ by more than 10\% within a single dataset. Is it a consequence of (1)?”*
>
> **A4**:The difference in accuracy across the models within a single dataset is due to the Boosting algorithm used, since all other variables were held constant (no. of WLs, type of WL, etc.). This is somewhat reasonable to expect since if the dataset is not realizable, then indeed we should expect realizable algorithms to perform poorly compared to their agnostic counterparts. The poor performance of the adaptive algorithm (Ada) is because it is has provably worse guarantees than its optimal counterparts (when the correct advantage parameter for the optimal algorithm is chosen). See [3] for the mistake-bounds of both OnlineMBBM and AdaBoost.OLM. It might be possible that a different type of WL might change which Boosting algorithm outperform one another. However, decision trees are a popular choice of WL and we stress that the point of our experimental section is to show that our OCO-based boosting algorithms are comparable to existing solutions while being much more computationally efficient.
>
> 1. N. Cesa-Bianchi and G. Lugosi. Prediction, Learning, and Games. Cambridge University Press, USA, 2006. ISBN 0521841089.
> 2. N. Brukhim, E. Hazan, S. Moran, I. Mukherjee, and R. E. Schapire. Multiclass boosting and the cost of weak learning. Advances in Neural Information Processing Systems, 34, 2021.
> 3. Y. H. Jung, J. Goetz, and A. Tewari. Online multiclass boosting. Advances in neural information
> processing systems, 30, 2017.

---

> ### Author Response · Authors · 2022-07-31
> **Response to Reviewer d9wh Part 2**
>
> **Q5**: *“It is difficult to make any conclusions from experimental results, since more expensive algorithms generally achieve higher accuracy. A “fair” comparison requires to assign equal computational budget to all methods (all of them have hyperparameters, which allow to trade-off accuracy and training time)."*
>
> **A5**: We thank the reviewer for this comment. We agree with the reviewer that there is no clear “winner” based on our experimental results. However, we would like to stress that the point of this section is to show that our OCO-based boosting algorithms are fast and comparable to existing algorithms.
>
> **Q6**: *“What parameters for Projected OGD are used?”*
>
> **A6**: We thank the reviewer for pointing out this missing detail. The only parameter needing to be set for projected OGD is the learning rate which we set to $\eta = \frac{\gamma}{\sqrt{N}}$ for both OCO-based boosting algorithms. This is the optimal learning rate for OGD up to constant factors for both boosting algorithms.  We will specify this in the final version of the paper.
>
> **Q7**: *“Why in OnlineMBBM the advantage parameter was tuned across [0.001, 0.01, 0.05, 0.1, 0.3]? Experimental part needs much more details.“*
>
> **A7**: We thank the reviewer for this comment and will add more experimental details in the final version. The range of gamma values we used to tune OnlineMBBM was also used by Jung et al. in [1]. We will clarify this in the final version of the paper. In addition, in our own experiments, we found that for gamma values larger than 0.30, OnlineMBBM performed significantly worse.
>
> **Q8**: *“Only seven quite small datasets are used. It would be useful to have also some average accuracy over all datasets and an alternative measure (e.g., cross-entropy).”*
>
> **A8**: In Boosting literature, these datasets and others of similar type have been used in prior works. For example, see [1, 2, 3, 4, 5]. We note that the guarantees of our boosting algorithms are with respect to the number of mistakes, which is why we used the accuracy as our measure of performance. We do not have any theoretical guarantees for other type of loss functions like cross-entropy. Also, note that boosting algorithms are not required to output a distribution over labels, but only a single label.
>
> **Q9**: *“In equations for expected regret (l. 69-70) and AWOL condition (ll. 155-156), $y_t / l_t$ should be different in the first and the second term of the differences, because $y_t / l_t$ depend on the expert/algorithm. This is the consequence of the environment adaptivity, and the use of the same notations for different variables indicates, conversely, the non-adaptive case.”*
>
> **A9**: We thank the reviewer for this comment, as it points out a subtle detail. In the equations for expected regret, the $y_t/l_t$ should actually be the same for the first and second terms in the differences. This is because the regret of the learner’s predictions is computed with respect to the performance of the best expert in hindsight for the SAME sequence the adversary has chosen for the learner. That is, the adversary chooses a sequence by playing a game with the learner, and the learner’s regret is measured with respect to the performance of the best fixed expert over this SAME sequence. This notation is standard and is also used on page 69 in [6], see Lemma 4.1.
>
> **Q10**: *“In ll. 67-68, the dimension of the standard simplex is k-1, not k”*
>
> **A10**:  We thank the reviewer for pointing this out and will make the appropriate changes in the final version.
>
> **Q11**: *“A small overview of OCO algorithms would be helpful for the reader, as OCO is a central part of the proposed algorithms.”*
>
> **A11**: We thank the reviewer for this suggestion. We do provide an overview of the OCO framework in the preliminaries section, but we will expand on this more in the final version.
>
> **Q12**: *“Caption for Figure 1: the dot after $p_2^\*$ appears at the beginning of the second line. (3) L187, second line, first term: factor 2 should be inside the bracket”*
>
> **A12**: We thank the reviewer for pointing out these issues and we will fix them in the final version.
>
> 1. Y. H. Jung, J. Goetz, and A. Tewari. Online multiclass boosting. Advances in neural information
> processing systems, 30, 2017.
> 2. A. Beygelzimer, S. Kale, and H. Luo. Optimal and adaptive algorithms for online boosting. In International Conference on Machine Learning, pages 2323–2331. PMLR, 2015.
> 3. S.-T. Chen, H.-T. Lin, and C.-J. Lu. An online boosting algorithm with theoretical justifications. arXiv preprint arXiv:1206.6422, 2012.
> 4. V. Kanade and A. Kalai. Potential-based agnostic boosting. Advances in neural information processing systems, 22, 2009.
> 5. Mukherjee, Indraneel, and Robert E. Schapire. "A theory of multiclass boosting." Journal of Machine Learning Research, 2013.
> 6.  N. Cesa-Bianchi and G. Lugosi. Prediction, Learning, and Games. Cambridge University Press, USA, 2006. ISBN 0521841089.

---

> ### Author Response · Authors · 2022-07-31
> **Response to Reviewer d9wh Part 3**
>
> **Q13**: *“The main setting considered in the paper is fully adversarial, not just agnostic: the environment adapts to the player. This entails a stronger condition on weak learners (see Definition 1) as compared to the same requirement in the non-adaptive case. At the same time, I believe, non-adaptive environment is more realistic for most practical cases. To what extent is the condition in Definition 1 more restrictive than that in the non-adaptive case? For example, are there generic examples for class of experts where the maximum of expected gain is at least positive? (as it required in ll. 518-520 of Supplementary?).”*
>
> **A13**: We thank the reviewer for this comment. Note that even if the adversary were non-adaptive in selecting the examples it presents to the Booster, the weak learning condition needs to hold under an adaptive adversary since they observe relabelled examples from the Booster. That is, the weak learners never interact with the adversary directly, but rather only through the Booster, which passes relabelled examples to them. So even if the adversary is non-adaptive to the Booster, the sequence of examples observed by the weak learner is not selected obliviously.
>
> We are not sure exactly how to quantify how much more restrictive the condition in Definition 1 is compared to the non-adaptive case. But, we show in Appendix D how to construct weak learners that satisfy Definition 1 in the adaptive setting. Note that even in the non-adaptive, oblivious setting, where the sequence of examples and labels is pre-specified before the game starts, the adversary can randomly label each example. Then, no class of experts can achieve positive maximum expected gain. Thus, even in a non-adaptive (oblivious) setting, Definition 5 in Appendix C (ll. 518-520) will not work, and one would need a condition potentially similar to Definition 1.
>
> **Q14**: *“I would consider a much simpler and interpretable condition for AWOL in Definition 1 without auxiliary variables $l_t$. We can just use 0-1 loss as the gain function instead of strange $\sigma_{y_t, l_t}$. Does the regret bound work for this version of Definition 1?”*
>
> **A14**: We thank the reviewer for this suggestion. The 0-1 loss is a loss function not a gain function. However, we have considered the 0-1 gain function (0 if incorrect and 1 if correct), but were unable to prove a regret bound.  In addition to the 0-1 gain function, we have tried many different gain functions. Two key requirements of any gain function is that (1) it should generalize the gain function when $k$ = 2 and (2) a randomly guessing competitor should achieve non-negative gain. Of the gain functions that satisfy property (1) and (2), the one presented in Definition 1 allowed us to prove a regret bound for Algorithm 1. This was actually the main difficulty of this work - finding an appropriate weak learning condition.
>
> **Q15**: *“Could you provide some intuition why do you want the environment to controls how much loss the weak learner suffers when it is incorrect?”*
>
> **A15**: We thank the reviewer for their comment. For the short answer, this was critical in allowing us to prove Theorem 1. Indeed, a subtle detail that enables the proof to work is the fact that we need to set $l_t^i = W_i(x_t)$ in order for the last equality in line 187 to hold. This requires knowledge of $W_i(x_t)$, which occurs only after $W_i(x_t)$ has played. In some sense, we can think of the Booster as the adversary for the weak learner, and it picks $l_t^i$ only after observing $W_i(x_t)$.
>
> **Q16**: *"Given that it the algorithms have the same regret bounds, are they equivalent in some sense?"*
>
> **A16**: We thank the reviewer for the question. Yes, Algorithm 1 does reduce to the algorithm of Brukhim et al. 2020 in the binary case. The weak learning condition in Definition 1 reduces directly. Furthermore, one can show that the exact same distribution $p_t^i$ over the labels is achieved by our algorithm and the one by Brukhim et al. in each round for each weak learner. We will further clarify this in the updated version.

---

### Meta-Review · Area_Chair_QSiX · 2022-08-23

**Recommendation:** Accept
**Confidence:** Certain

**Metareview:**

The reviewers agree that this is a solid contribution. Please do revise the paper according to the reviewers comments and the discussion.

**Award:**

No

---

### Decision · Program_Chairs · 2022-09-14

Accept